# The RNA-binding protein YBX1 regulates epidermal progenitors at a posttranscriptional level

Eunjeong Kwon[1], Kristina Todorova[1], Jun Wang[1], Rastislav Horos[2], Kevin K. Lee[1], Victor A. Neel[3], Gian Luca Negri[4,5], Poul H. Sorensen[4,5], Sam W. Lee[1,6], Matthias W. Hentze[2] & Anna Mandinova[1,6,7]

The integrity of stratified epithelia depends on the ability of progenitor cells to maintain a balance between proliferation and differentiation. While much is known about the transcriptional pathways underlying progenitor cells' behavior in the epidermis, the role of posttranscriptional regulation by mRNA binding proteins—a rate-limiting step in sculpting the proteome—remains poorly understood. Here we report that the RNA binding protein YBX1 (Y-box binding protein-1) is a critical effector of progenitors' function in the epidermis. YBX1 expression is restricted to the cycling keratinocyte progenitors in vivo and its genetic ablation leads to defects in the architecture of the skin. We further demonstrate that YBX1 negatively controls epidermal progenitor senescence by regulating the translation of a senescence-associated subset of cytokine mRNAs via their 3′ untranslated regions. Our study establishes YBX1 as a posttranscriptional effector required for maintenance of epidermal homeostasis.

[1] Cutaneous Biology Research Center, Massachusetts General Hospital and Harvard Medical School, Charlestown, MA 02129, USA. [2] European Molecular Biology Laboratory (EMBL), Meyerhofstrasse 1, 69117 Heidelberg, Germany. [3] Department of Dermatology, Massachusetts General Hospital and Harvard Medical School, Boston, MA 02114, USA. [4] Department of Pathology and Laboratory Medicine, University of British Columbia, Vancouver, BC V5Z 1L3, Canada. [5] Department of Molecular Oncology, British Columbia Cancer Research Centre, Vancouver, BC V5Z 1L3, Canada. [6] Broad Institute of Harvard and MIT, 7 Cambridge Center, Cambridge, MA 02142, USA. [7] Harvard Stem Cell Institute, 7 Divinity Avenue Cambridge, Cambridge, MA 02138, USA. Correspondence and requests for materials should be addressed to A.M. (email: amandinova@mgh.harvard.edu)

Control of stem cell fate, self-renewal, and commitment to programmed differentiation or death is fundamental for tissue homeostasis, regeneration, and aging[1, 2]. In recent years, the epidermis with its multiple cell lineages, high degree of turnover, and ability to withstand continuous exogenous injury has become a paradigm for studying stem cell homeostasis[3]. Epidermal stem cells have both quiescent and actively cycling populations[4, 5]. Upon activation, stem cells enter a transitory state of rapid proliferation, followed by exit from the cell cycle and commitment to differentiation[1]. During this process, progenitor cells need to be protected from undergoing senescence, which can be a default state for rapidly proliferating cells[6]. A breakdown in the mechanisms controlling the self-renewal process have been linked to a variety of common skin disorders[7].

Attempts to dissect the molecular pathways governing epidermal self-renewal have largely focused on transcriptional and epigenetic control of differentiation-related genes. By contrast, posttranscriptional regulation of epidermal stem cell biology by RNA-binding proteins (RBPs) is largely unexplored in spite of its general importance for sculpting the cellular proteome[8, 9]. In the field of stem cell biology, the highly conserved RBP Lin28 has emerged as a key factor that defines "stemness" in several tissue lineages[10]. While Lin28 expression is physiologically restricted to embryonic tissues, its misexpression in the adult skin affects epidermal stem cell function with promotion of epidermal hair growth and altered tissue regeneration[10]. Another member of the same family of cold-shock domain-containing RBPs, YBX1, is expressed in embryonic tissues but is also normally present in the adult epidermis[11]. YBX1 has been reported to modulate the overall levels of protein synthesis and to directly enhance the translation of prominent cancer stem cell factors such as Twist, Snail, Myc, and HIF1α, whereas it can inhibit the translation of oxidative phosphorylation-related proteins in cervical cancer cells[12–15]. These reports point to YBX1 as a regulator of cellular proliferation, the metastatic potential of cancer cells, and a determinant of cancer stem cell function[16–18]. In epidermal stem cells, YBX1 partners with the RNA helicase DDX6 and binds the 3′ untranslated regions (UTRs) of regulators of self-renewal such as CDK1 and EZH2[19] to facilitate their translation.

Cellular senescence and aging are associated with a decreased ability of tissues to regenerate, frequently associated with impaired stem cell function[20, 21]. Age-associated imbalances in cytokine signaling in keratinocytes induce senescence, lower the ability of the epidermis to tolerate stress, and inhibit stem cell function[4]. To maintain epidermal homeostasis, suppression of senescence is likely to be required for all epidermal cells, whether quiescent, actively proliferating, or undergoing differentiation. The underlying mechanisms of senescence control are therefore important to be uncovered both in normal and pathological conditions.

Senescent cells initiate a complex program called the senescence-associated secretory phenotype (SASP)[22, 23]. Precise mechanisms of molecular control of SASP remain unclear although alterations in cytokine abundance are generally affected at the level of gene transcription[24]. Specific cytokine signaling has recently been suggested to inhibit epidermal stem cell function[4], but a direct link to SASP has not been established yet.

Here we report the critical role of the RBP YBX1 in the maintenance of epidermal progenitor cells in vivo and in vitro and uncover its function as a translational repressor of cytokines involved in the promotion of senescence.

## Results

**YBX1 is an RBP enriched in epidermal progenitor cells.** Epidermal progenitor cell function is regulated transcriptionally by several prominent transcription factors[25–28] and by epigenetic effectors such as DNMT1 and EZH2 that assist in maintaining keratinocyte stem and progenitor cell function[29–31]. However, it is now recognized that the cellular proteome is heavily controlled at the posttranscriptional level with RBPs taking a leading role in this process[32].

To identify RBPs participating in maintaining epidermal homeostasis, we performed poly(A)-mRNA interactome capture (Fig. 1a) as previously described[33–35]. The efficiency of poly(A)-mRNA pull-down by oligo(dT) beads was confirmed by enrichment of captured mRNAs (18S rRNA, β-actin and hypoxanthine phosphoribosyltransferase 1, HPRT) over exogenous luciferase mRNA (spiked in) as measured by quantitative reverse transcription-polymerase chain reaction (qRT-PCR) analysis of the recovered RNA fractions (Fig. 1b). In vivo cross-linking of proteins in extremely close proximity to mRNAs resulted in efficient capture of multiple polypeptides. Stringent washing steps enabled effective removal of the abundant proteins not directly bound to poly(A)-RNAs as shown by silver gel staining (Fig. 1c, lanes 5 and 6). The "captured" samples showed profoundly different staining pattern as compared to the input (lanes 1–4), reflecting the specificity and selectivity of the method (Fig. 1c). Western blot analysis of the oligo(dT) selected samples detected the bona fide RBP polypyrimidine tract-binding protein 1 (PTBP1) exclusively in the UV-cross-linked samples, while abundant cellular proteins such as histone H3 (H3) was present only in the input sample without contamination of the captured material (Fig. 1d), confirming the high selectivity and specificity of this approach.

We performed three biologically independent experiments using primary epidermal cells derived from different adult donors and observed surprisingly low variability among individuals (Supplementary Fig. 1A). From the three biological replicates, liquid chromatography-tandem mass spectrometry (LC-MS/MS) analysis identified a total of 673 proteins, 348 of which were quantified and enriched in UV-cross-linked samples from at least two replicates (Supplementary Data 1 and Supplementary Fig. 1B). Among those, the majority of the captured proteins were classified as known RBPs and modulators of RNA biology using Gene Ontology (GO) (Supplementary Data 1). When we examined the global expression profile of these mRNA interactors during epidermal progenitor differentiation using published transcriptome data[36], we identified 61 RBPs from the dataset with high level of expression (greater than 30 FPKM from the publically available RNA-seq dataset) in progenitors, which are substantially downmodulated upon keratinocyte commitment to differentiation (Fig. 1e, Supplementary Data 2). Close examination of the RBPs with highest degree of downmodulation during differentiation (top 30 classical RBPs: Supplementary Data 2) identified YBX1 as a potentially important player in epidermal progenitor function due to its role as a regulator of epidermal stem cells[19] and epithelial cancer stem cells[13] as well as keratinocyte differentiation through an interaction with p63[11].

Analysis of the oligo(dT) captured mRNA-protein complexes in UV-cross-linked proliferating epidermal progenitors by western blot confirmed the enrichment of YBX1 at the poly(A)-tagged fraction of mRNAs in this cell population (Fig. 2a). To further confirm the RNA affinity of YBX1 in epidermal progenitors, we immunoprecipitated the protein from UV-irradiated primary human keratinocytes, and radioactively labeled the cross-linked transcripts by polynucleotide kinase (PNK) treatment after gradual digestion with RNase. Analysis of the immunoprecipitate by gel electrophoresis and autoradiography revealed the cross-link-dependent co-purification of RNA with YBX1 (Fig. 2b). When primary human keratinocytes were grown in "progenitor state" preserving conditions (less than 60% confluence) and

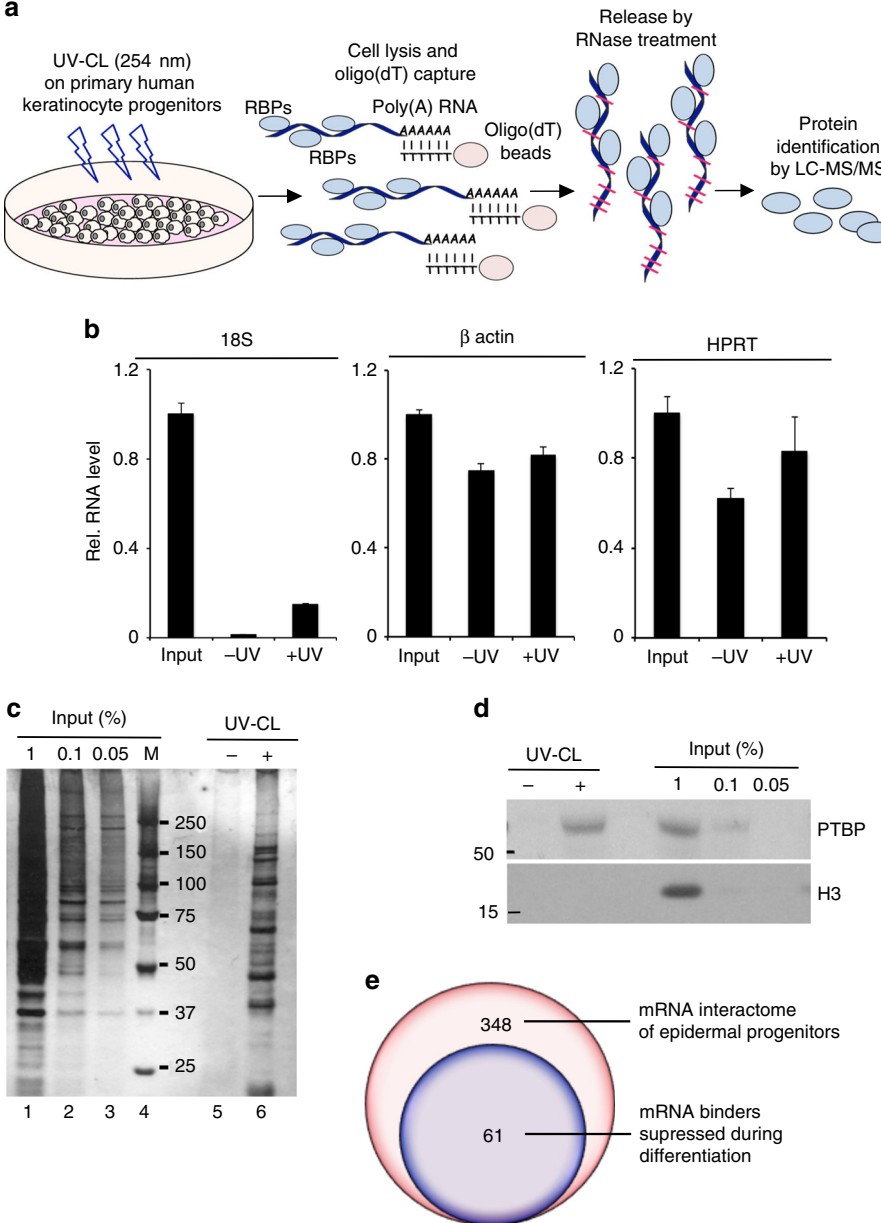

**Fig. 1** Identification of RNA binding proteins (RBPs) in primary human keratinocyte progenitors. **a** Schematic of the mRNA interactome-capture approach in primary human keratinocyte cultures. Three independent biological replicates were performed with pooled primary cultures of different donors. Cells were UV-cross-linked or nonirradiated and the bound to poly(A)-tagged mRNAs polypeptides were enriched and eluted from the oligo(dT) beads and further identified by LS-MS/MS. **b** Levels of 18s rRNA, β-actin, and hypoxanthine phosphoribosyltransferase 1 (HPRT) mRNAs in eluates from oligo(dT) captured samples were evaluated by qRT-PCR. Exogenous poly(A)-tailed luciferase mRNA was "spiked" into the lysates and used as an internal control. Error bars represent mean SD, $n = 3$. **c** Eluted proteins were visualized by SDS-PAGE and silver staining. **d** Western blotting on the eluted samples for levels of PTBP and histone H3. **e** RBPs expression profile in human keratinocytes during differentiation ([36], data accessible at NCBI GEO database, accession GSE58161)

subsequently prompted to undergo spontaneous differentiation by exceeding 80% confluence[37], both transcript and protein levels of YBX1 were downregulated, while the differentiation markers Keratin 10 and Involucrin were upregulated as expected (Fig. 2c, d). We also found that YBX1 expression is progressively downmodulated in the epidermis during late embryonic development in mice in vivo (Fig. 2e). In the epidermis, undifferentiated progenitor cells residing in the basement membrane-bound basal layer undergo cell cycle arrest, outward migration, and terminal differentiation to generate cutaneous permeability barrier. Consistent with a role in epidermal progenitor function, YBX1 was mainly confined to cells of the basal layer of adult human epidermal tissue (Fig. 2f). Similarly, in the mouse

epidermis, expression of YBX1 was detected in the basal layer of the interfollicular epidermis, the secondary hair germ and the outer layer of the sebaceous glands (Fig. 2g). Thus, our findings indicate that the RBP YBX1 is expressed in epidermal progenitor-containing cell populations and is lost during differentiation.

**In vivo loss of function of YBX1 leads to epidermal defects.** The downregulation of YBX1 during epidermal differentiation suggested the possibility that YBX1 may impact on epidermal progenitor function. To explore this, we studied the effects of genetic ablation of YBX1 in the mouse. While YBX1 haploinsufficiency (YBX1$^{+/-}$) does not lead to any visible phenotypic

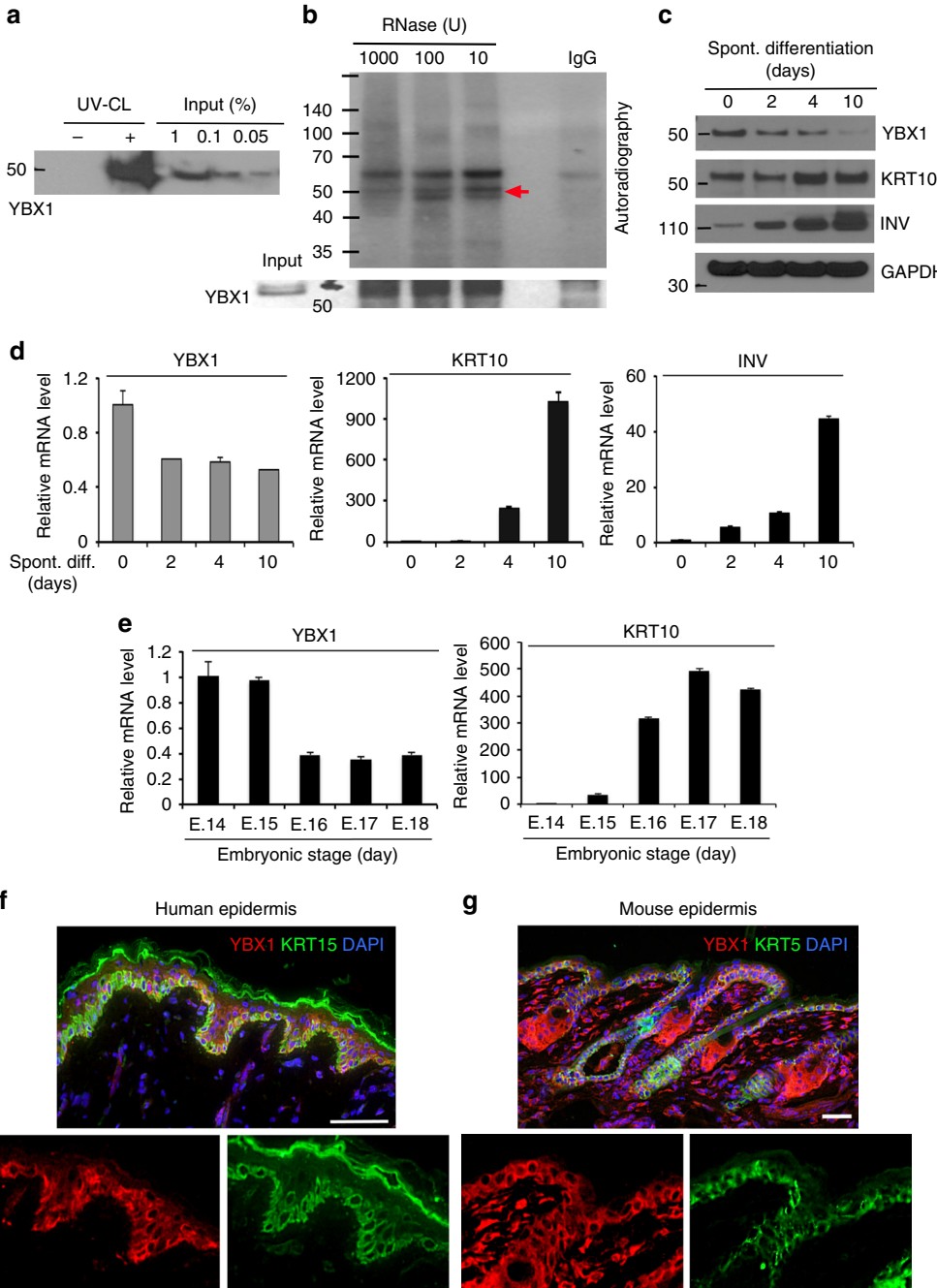

**Fig. 2** The expression of the RBP YBX1 is decreased during keratinocytes commitment to differentiation and at late stages of embryonic development. **a** Eluates from a small scale oligo(dT) experiment with proliferating epidermal progenitors were analyzed by western blotting for YBX1 protein levels. **b** RNA-binding of YBX1 in human epidermal progenitors. Endogenous YBX1 was immunoprecipitated, and UV-cross-linked transcripts were trimmed by incubation with an RNase I, labeled with polynucleotide kinase and [γ-$^{32}$P]ATP and the complexes were separated by denaturing gel electrophoresis prior to visualization by autoradiography (the red arrow points to YBX1 associated RNAs). The efficiency of YBX1 immunoprecipitation was confirmed by western blotting. **c** Protein levels of YBX1, Keratin 10 (KRT10), Involucrin (INV), or GAPDH (as a loading control) in spontaneously differentiating human keratinocytes (for 10 days) were analyzed by western blotting. **d** YBX1, KRT10, and INV mRNA levels in primary cultures of human keratinocytes induced to spontaneously differentiate for 10 days were measured by quantitative RT-PCR with 36B4 mRNA levels as a control. Error bars represent mean SD, $n = 3$. **e** Levels of YBX1 and KRT10 mRNA in the mouse epidermis during embryonic days 14 to 18 (E.14–E.18) were quantified by qRT-PCR using mRNA levels of 36B4 as an internal control. Error bars represent mean SD, $n = 3$. **f, g** YBX1 expression in vivo in the human (**f**) and mouse (**g**) skin. Immunofluorescence and confocal microscopy were used to detect YBX1 (shown in red) and the basal layer markers Keratin 15 (KRT15 for human epidermis) and Keratin 5 (KRT5 for mouse epidermis), shown in green. The nuclei were stained with DAPI in blue. Magnified images generated to present single channels are shown for better resolution. Shown images are representative of at least $n = 7$. Scale bars for the human epidermis are 65 μm and for the mouse epidermis 32 μm

defects in mice, homozygous deletion of YBX1 (YBX1$^{-/-}$) results in high rates of late stage embryonic lethality. Some embryos survive until embryonic days 15 to 16 (E.15–16) but a very limited number are alive at birth[38], allowing us to analyze the morphology of the epidermis and the hair follicles at the time of completed development and stratification between E.15 and E.16. YBX1$^{-/-}$ embryos at E.16 appeared smaller than wild-type (WT)

littermates (Supplementary Fig. 2A-D). Histological analysis of their epidermis revealed markedly reduced thickness of the epidermis and significantly (as determined by unpaired $t$-test) delayed onset of the placodes to hair follicle transition as measured by the numbers of developed hair follicles (Fig. 3a) suggesting defects in the hair follicle stem cells and/or their actively cycling progeny[39]. Further, Ki67 labeling showed substantially

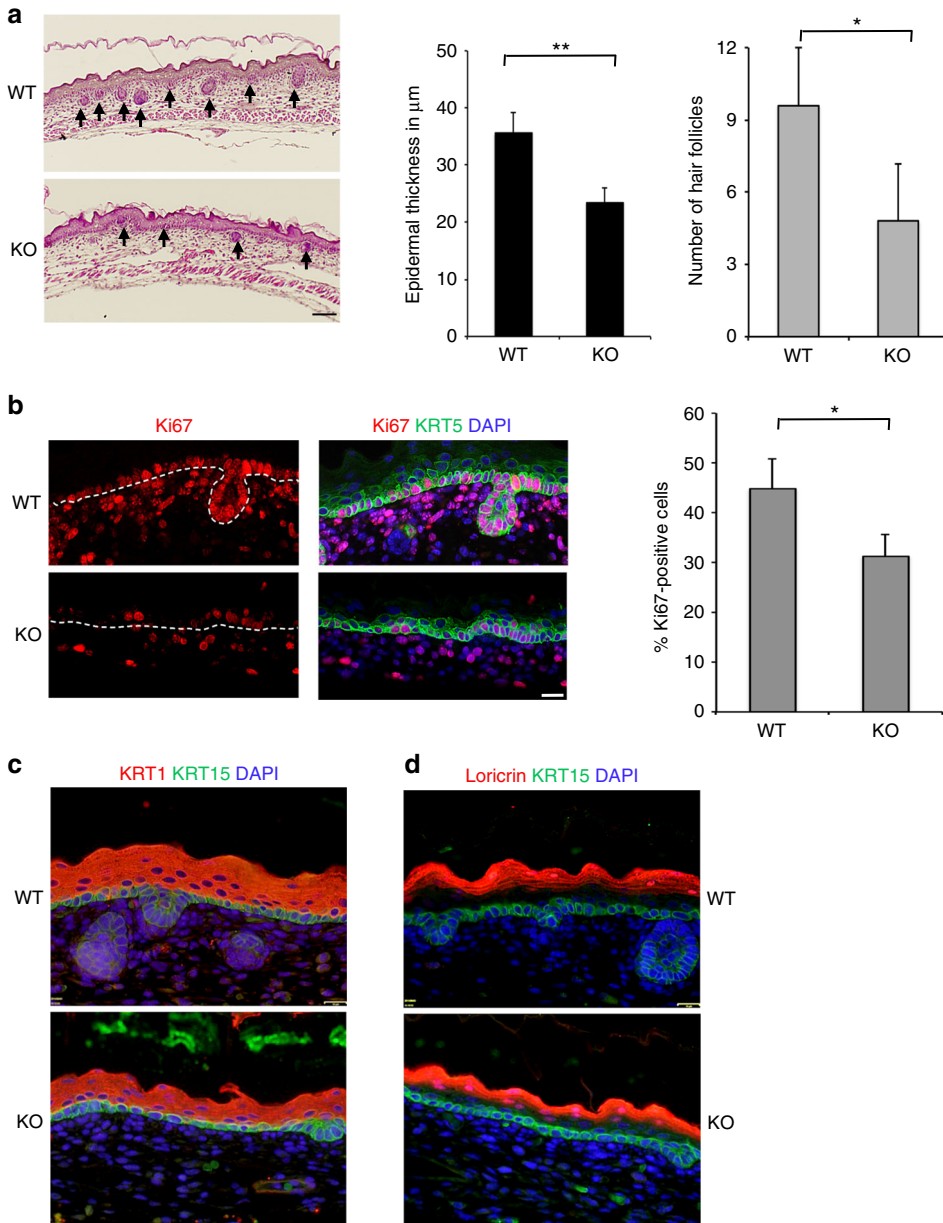

**Fig. 3** Genetic ablation of YBX1 in the mouse epidermis results in reduced number of hair follicles and diminished proliferative capacity of the interfollicular epidermis. **a** Histological analysis of the mouse epidermis at the final stages of embryonic development. Embryos at embryonic day 16 (E.16) from wild type (WT) and whole body YBX1$^{-/-}$ (KO) mice were collected, formalin fixed, paraffin embedded, and H&E stained. Arrowheads point to the hair follicles. Epidermal thickness was measured using ImageJ on five different samples per phenotype at five different positions along the epidermis and values are presented in μm as average ± SD, **$p < 0.01$. Hair follicles were counted on five different samples per phenotype and values are average ± SD, *$p < 0.05$, unpaired $t$-test. Scale bar = 64 μm. **b** The proliferative capacity of the epidermis was detected by immunofluorescence staining for Ki67. Mouse embryos at embryonic day 16 (E.16) from WT and KO mice were collected and stained with Ki67 (red), Keratin 5 (KRT5, green), and DAPI (blue). The white dotted line marks the basement membrane underlying the epidermis. Results were visualized by fluorescence microscopy and the percent of Ki67-positive cells (compared to all KRT5-positive basal cells) was determined using ImageJ on four different samples per phenotype. Values are average ± SD, *$p < 0.05$, unpaired $t$-test. Scale bar = 16 μm. **c, d** Mouse embryonic epidermis from WT and KO animals (at day 16 as in **b**) was stained for the differentiation markers KRT15 (basal layer: in green), KRT1 in **c** (suprabasal layer: in red) and Loricrin in **d** (granular layer: in red). DAPI staining was used to visualize the nuclei. Shown images are representative of $n = 6$. Scale bar = 16 μm

decreased proliferative capability of the interfollicular epidermis and the underlying dermal tissue, which is indicative of functional failure of proliferating epidermal and dermal progenitors (Fig. 3b). Immunofluorescence staining revealed normal epidermal differentiation and stratification, with similar expression of Keratin 15 (KRT15) and Keratin 5 (KRT5) in the basal layer, Keratin 1 (KRT1) in the suprabasal layer and loricrin in the granular layer of the epidermis of wild-type YBX1$^{+/+}$ (WT) and

knockout YBX1$^{-/-}$ (KO) mice (Fig. 3c, d; Supplementary Fig. 2E, F).

**Depletion of YBX1 reduces the cycling epidermal progenitors.** Consistent with a function of YBX1 in maintaining the epidermal progenitor state, YBX1 loss in human primary keratinocytes impaired clonogenic growth as compared to controls (Fig. 4a).

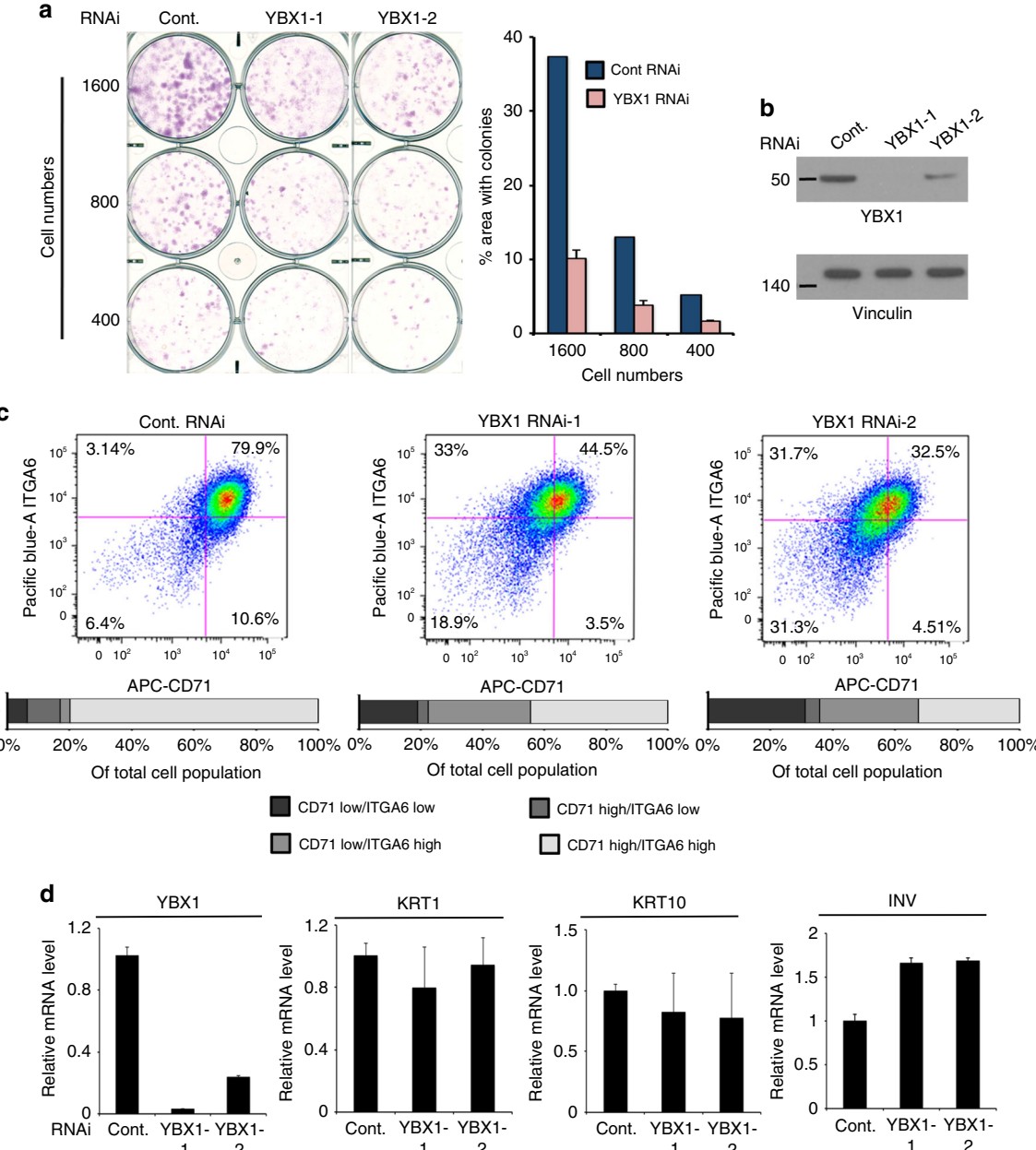

**Fig. 4** YBX1 depletion in primary human keratinocytes results in epidermal progenitor defects without affecting commitment to differentiation. **a** Colony formation capacity of primary human keratinocytes transfected either with a control siRNA or two different YBX1 siRNAs was assessed by a clonogenic assay; 400, 800, or 1600 cells per well were seeded in 6-well plates and incubated for 10 days. Cells were fixed and stained using Sulforhodamine B (SRB). % area with colonies was calculated using ImageJ. Average and SDs were calculated based on the two different siRNAs results. **b** YBX1 protein levels in keratinocytes transfected with control or YBX1 siRNAs (from **a**) were assessed using western blotting, and vinculin levels served as a loading control. **c** YBX1 depletion reduces the population of actively cycling epidermal progenitors: passage 2 of primary human keratinocytes pooled from three different donor cultures were transfected with either control siRNA or two different YBX1 siRNAs and sorted for expression of CD71 and Integrin-α6 (ITGA6) by a flow cytometer after staining with APC-conjugated anti-CD71, and Pacific blue-conjugated anti-ITGA6 antibodies. Quantification of the separate populations of cells is presented in the lower panel as % of the whole population. **d** Control and YBX1 siRNA transfected human keratinocytes were harvested 4 days after transfection and mRNA levels of the differentiation markers KRT1, KRT10, and INV were quantified by qRT-PCR using 36B4 mRNA as an internal control. Error bars represent mean SD, $n = 3$

The human epidermis contains quiescent and actively cycling pools of basal cells, which contribute to self-renewal and tissue homeostasis[1]. While populations of both quiescent and cycling basal cells are commonly characterized by high levels of the cell surface marker integrin α6 (ITGA6), CD71/transferrin receptor is used to separate both populations into quiescent (ITGA6bri/CD71dim) and actively cycling (ITGA6bri/CD71bri)[40]. To analyze the effect of YBX1 on epidermal progenitors, we sorted human primary keratinocytes with and without YBX1 depletion and demonstrated that decreased expression of YBX1 resulted in strongly diminished numbers of actively cycling (ITGA6bri/CD71bri) epidermal progenitors (Fig. 4b, c), which correlated with

the phenotype observed in vivo (Fig. 3a) as well as with decreased clonogenic and regenerative capabilities in vitro. Notably, we observed a more pronounced decrease in the CD71 labeling as compared to ITGA6 upon YBX1 downmodulation indicating reduction in keratinocytes proliferation. Similarly to the normal stratification pattern observed in vivo, the mRNA levels of the differentiation markers KRT1, KRT10, and INV in grown over 80% confluence primary cultures of human keratinocytes were not affected upon YBX1 depletion in vitro. (Fig. 4d) These findings indicate that while YBX1 maintains epidermal progenitor homeostasis, it does not directly target the transcriptional control of keratinocytes commitment to differentiation.

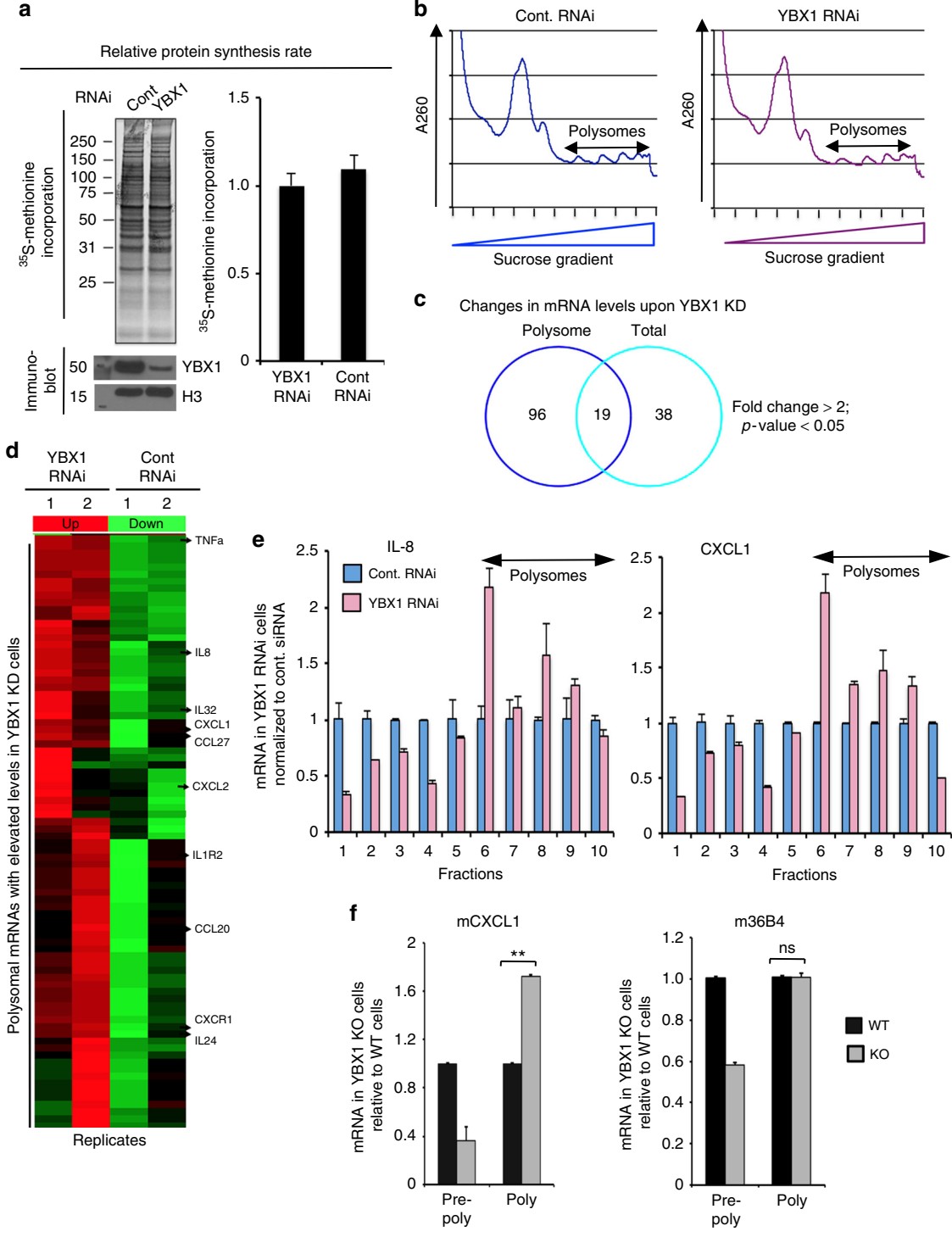

**YBX1 binds to cytokine mRNAs in epidermal progenitors**. To investigate the mechanisms underlying the effects of YBX1 depletion on epidermal progenitors, we performed deep RNA-seq analysis of control versus YBX1-depleted proliferating epidermal cells. [35]S-Methionine incorporation assay revealed no effect of YBX1 depletion on global translation (Fig. 5a). Since YBX1 can affect both transcription and translation, we analyzed both total mRNA and polysome-associated mRNAs using sucrose density gradient sedimentation (Fig. 5b). Deep RNA-seq analysis based on duplicate samples showed that 57 genes were substantially modulated in YBX1 siRNA treated cells at the level of total mRNA as compared to control siRNA cells, while 115 genes exhibited changes in polysomal mRNA, respectively (Fig. 5c). In line with our previous observations, YBX1 depletion did not have a considerable effect on genes related to epidermal differentiation (Supplementary Data 3; diff. expressed genes in total fractions). Considering the predominantly cytosolic localization of YBX1 in the epidermis (Fig. 2f, g), we hypothesized that most of the observed transcriptional changes are secondary to changes in YBX1-mediated translational regulation of specific mRNA subsets. To identify such translational targets of YBX1, we analyzed transcripts that were equally expressed within total RNA populations of control and YBX1 siRNA cells but were selectively modulated in the polysomal mRNA fractions of YBX1-depleted cells (Fig. 5d; Supplementary Data 4). Further examination revealed that out of the 96 affected polysomaly associated mRNAs, 39 were most discriminatory in expression in polysomal versus total mRNA (Supplementary Data 4). Surprisingly, GO analysis indicated that the YBX1 driven global changes in the actively translating portion of the transcriptome corresponded mainly to those genes associated with chemokine and cytokine signaling (Supplementary Data 5). Consistent with an established function of YBX1 as a translational repressor, the majority of transcripts detected in the polysomal fraction of YBX1 knockdown (KD) cells were increased as compared to control cells. In agreement with previously published data[13], we also identified a small set of YBX1-positive targets, among which, mediators of epithelial–mesenchymal transition (EMT) such as SHIP1, GPR124, and AGR2 were present (Fig. 5d; Supplementary Data 4). Next, qRT-PCR confirmed that KD of YBX1 in proliferating epidermal cells indeed leads to increased polysomal association of a defined subset of cytokines including CXCL1 and its structural homolog IL-8 (Fig. 5e), as well as CCL20, IL-24, and TNFα (Supplementary Fig. 3). We were also able to isolate limited amounts of primary mouse keratinocytes derived from WT and YBX1 KO embryos at E.15–16 and observed similar increase of mouse CXCL1 in the polysomal fractions of YBX1 KO keratinocytes as compared to the WT cultures (Fig. 5f). This effect was specific only for certain mRNAs without affecting global translation (Fig. 5f). The total mRNA levels of both positive and negative YBX1 targets remained similar between WT and YBX1 KD keratinocytes arguing in favor of a predominantly translational mechanism of control.

**YBX1 suppresses the translation of IL-8 and CXCL1**. Next, we investigated whether YBX1 is able to bind directly to the mRNAs of its targets by RNA immunoprecipitation (RNA-IP). To this end, endogenous YBX1 was immunoprecipitated from whole cell lysates of proliferating keratinocytes using an anti-YBX1 antibody (MBL, Japan). The co-immunoprecipitated RNA was purified and the enrichment levels of CXCL1, IL-8, and CXCL2 transcripts were assessed by qRT-PCR. Previously established binding of YBX1 to its own mature mRNA was used as a positive control[41,42], while mitochondrially encoded mRNAs served as negative controls[14] (Fig. 6a). Earlier examples of suppression of cytokine translation were found to be primarily mediated by a 3′ UTR-dependent mechanism[43]. Therefore, we sought to establish whether YBX1 uses the 3′UTR of its target cytokines to modulate their translation. We used two reporter constructs containing a firefly luciferase gene (luc) cloned in front of the 3′UTRs of the IL-8 and the CXCL1 mRNAs, respectively. In the first set of experiments, we found that in mouse embryonic fibroblasts (MEF) genetic ablation of YBX1 resulted in elevated levels of the 3′UTR IL-8-luc reporter signal as compared to the control WT MEFs (Fig. 6b). A luc construct with a minimal 3′UTR served as a negative control and did not show a difference of translation efficiency between WT and YBX1 KO cells (Fig. 6b). Next we transfected the luc construct containing the 3′UTR of CXCL1 into YBX1 KO MEF cells and showed that transient expression of WT YBX1 was able to repress the luciferase signal from the 3′ UTR CXCL1-luc reporter while a YBX1 mutant lacking its RNA-binding cold-shock domain failed to do so[44] (Fig. 6c). To further examine the 3′UTRs of the YBX1 target mRNAs, we performed a bioinformatics analysis for common consensus motifs. It has been previously established that cytokine production at the level of protein biosynthesis is primarily suppressed via interaction of RBPs with AU-rich elements (ARE) located in their 3′UTRs[45]. Analysis of Motif Enrichment (AME)[46] showed a substantial enrichment for 5 out of 6 ARE motifs on the 3′UTRs sequence of 13 out of the 23 downregulated transcripts (Supplementary Fig. 4). Among those, we noted the IL-8 mRNA to contain numerous of these consensus sequences in its 3′UTR and for the next set of experiments we generated 3′UTR-luc IL-8 deletion mutants lacking sequential clusters of AREs (Fig. 6d). The reporter activity of the mutants was than compared to the full-length construct upon co-transfection with WT YBX1 cDNA into YBX1 KO MEFs. The M3 IL-8 3′UTR-luc mutant lacking the first and the second ARE exhibited a markedly diminished capacity to respond to YBX1-mediated suppression of translation when

**Fig. 5** YBX1 modulates the translation of downstream target mRNAs including a subset of chemo- and cytokines. **a** Human keratinocytes were incubated with [[35]S]methionine-containing medium for 1 h and [[35]S]methionine incorporation into polypeptides was detected by SDS-PAGE and TCA precipitation. **b** A representative polysome profile of primary human keratinocytes transfected with either control or YBX1 siRNA: polysomal fractions of the mRNA from primary human keratinocytes transfected with the respective siRNAs for 4 days were isolated by 10–45% sucrose gradients. **c** High-resolution RNA-seq analysis of total and polysomal fractions of mRNA from control versus YBX1-depleted proliferating epidermal cells was performed using two primary cultures of keratinocytes isolated from independent donors: Venn diagram of mRNAs in total versus actively translating polysomal fractions. **d** Heat map of polysome-associated mRNAs with elevated levels in cells with siRNA-mediated knockdown of YBX1. Selected chemo- and cytokines are indicated in the polysome-associated fractions (two biological replicates are shown). **e** Polysome-associated mRNAs were analyzed in primary cultures of human keratinocytes transfected with control or YBX1 RNAi 4 days after transfection: ribosomal subunits and polysomes were fractionated by a sucrose gradient and monitored with continuous $A_{260}$ detection as in **b**. mRNA was extracted from each fraction and fold changes of mRNA levels for IL-8 and CXCL1 in YBX1-depleted cells were analyzed relative to controls by qRT-PCR after normalization for 18S mRNA. Fractions 1–5 represent light, pre-polysomal mRNAs and fractions 6–10 contain the heavy polysome-associated mRNAs. Error bars represent mean SD. **f** Total mRNA from WT and YBX1 KO mouse primary keratinocytes (from embryos at E.16) was isolated and separated into pre-polysomal and polysomal fractions. Relative changes in the KO versus WT mRNA levels of mouse CXCL1 (mCXCL1) and 36B4 (m36B4) in the pre-polysomal and polysomal fractions were quantified by qRT-PCR using 18S as a loading control. Error bars represent mean SD, *$p < 0.05$; ns: $p > 0.05$, unpaired $t$-test

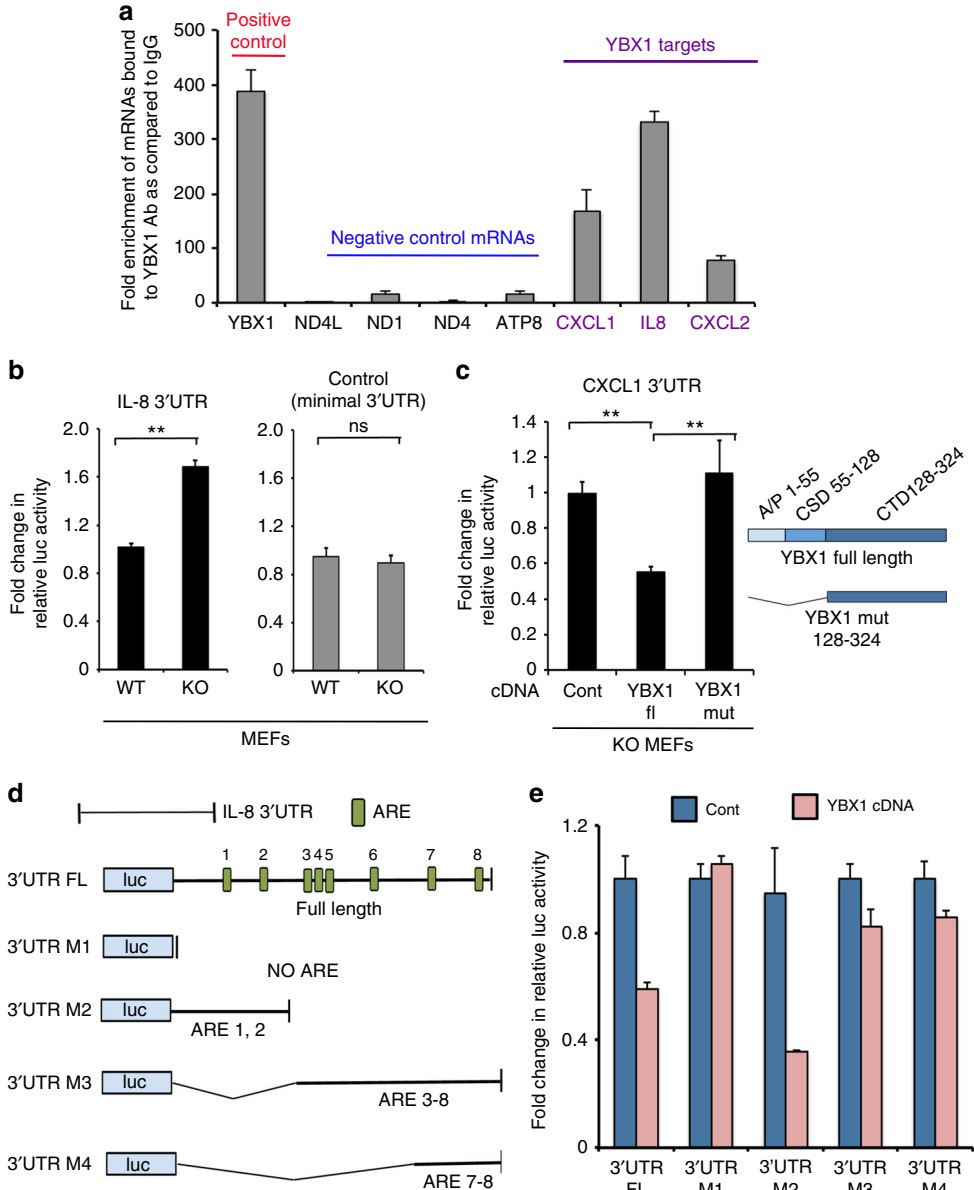

**Fig. 6** YBX1 binds to the 3'UTRs of transcripts encoding cytokines and chemokines. **a** RNA immunoprecipitation in proliferating cultures of human keratinocytes was performed using control IgG or YBX1-specific antibody. RNAs from each immunoprecipitate were purified and quantified by qRT-qPCR using primers for the indicated targets. Fold enrichment levels were calculated relative to the IgG control after normalization to the respective inputs. YBX1 mRNA was used as a positive control, the mitochondrial mRNAs encoding NDL4, ND1, ND4, and ATP8 served as negative controls and IL-8, CXCL1, and CXCL2 were detected as positive targets. Error bars represent mean SD, $n = 3$. **b** WT and YBX1 KO MEFs were transfected with a 3'UTR IL-8 or a control luc reporter construct containing a minimal 3'UTR sequence; 48 hours later, the cell lysates were harvested and the luciferase activity was measured using EnVision plate reader. Error bars represent mean SD, $n = 3$, *$p < 0.05$; ns: $p > 0.05$, unpaired $t$-test. **c** YBX1 KO MEFs were co-transfected with a 3'UTR CXCL1 reporter construct and wild type YBX1 or a deletion YBX1 mutant lacking the RNA-binding cold-shock domain; 48 h later, the cell lysates were harvested and the luciferase activity was measured using EnVision plate reader. Error bars represent mean SD, $n = 3$, **$p < 0.01$; *$p < 0.05$, unpaired $t$-test. **d** The 3'UTR of human IL-8 contains several putative AREs (AU rich elements). Deletion mutants (M1–4) lacking certain AREs (as indicated) were generated by restriction digestion of the full-length (FL) 3'UTR construct. **e** The FL IL-8 3'UTR construct and the respective deletion mutants (M1–4) were co-transfected together with control or YBX1 cDNA in YBX1 KO MEF, and 48 h later, the cell lysates were harvested and the luciferase activity was measured using EnVision plate reader. Error bars represent mean SD, $n = 3$

compared to the full-length construct and the remaining mutants (Fig. 6e). Together these results indicate that YBX1 binds to the 3' UTR of target transcripts such as IL-8 and CXCL1 to translationally repress their expression. Although these studies also implicate certain AREs in the 3'UTRs of the targeted transcript for YBX1-mediated effect, further studies are needed to determine each specific UTR sequence and tertiary structure bound by YBX1.

**YBX1 protects epidermal progenitors from senescence.** We next sought to establish the functional significance of YBX1 driven modulation of cytokine biosynthesis and showed that siRNA-mediated KD of YBX1 expression in primary human keratinocytes resulted in a considerable increase of CXCL1 and IL-8 production as compared to control RNAi (Fig. 7a, b). YBX1 KD was also accompanied by a decrease in the number of cells when compared to control cultures seeded under equal conditions

(Fig. 7c). This effect was specific to the RNAi-mediated suppression of YBX1 since cells infected with an adenoviral construct encoding an RNAi resistant YBX1 recovered the difference in cell numbers (Supplementary Fig. 5).

It has been previously shown that the subset of cytokines negatively regulated by YBX1 in keratinocytes belongs to a group with a well-established ability to promote a complex, senescence-related phenotype termed the SASP[22, 23, 47]. This subset of cytokines is able not only to promote intrinsic senescence but also exerts a paracrine effect, transmitting the phenotype to bystander cells[48]. Premature entry into senescence would explain the observed stem cell defects associated with YBX1 deficiency in vitro such as alterations of the clonogenic activity and depletion of CD71-positive cells (Fig. 4). Similarly, decline in the

epidermal thickness, decrease of the number of developed hair follicles and Ki67-positive cells in vivo at E.16 in YBX1 KO mice support the presence of a senescent phenotype in YBX1-depleted tissues (Fig. 3). Accordingly, cell cycle analysis of primary human keratinocytes depleted of YBX1 expression detected increased number of cells arresting in the G2/M phase (Fig. 7d). KD of YBX1 also increased the activity of senescence-associated-β-Galactosidase (SA-β-Gal) (Fig. 7e). Furthermore, p21$^{CIP}$, another senescence marker was similarly elevated in cells with YBX1 KD (Fig. 7f). In agreement with these findings in vitro, we observed an increase in senescence-associated heterochromatin foci in vivo, visualized with fluorescent staining for trimethylated histone H3 lysine 9 (H3K9me3) in the basal layer (containing KRT15-positive cells only) of the epidermis of YBX1 knockout animals,

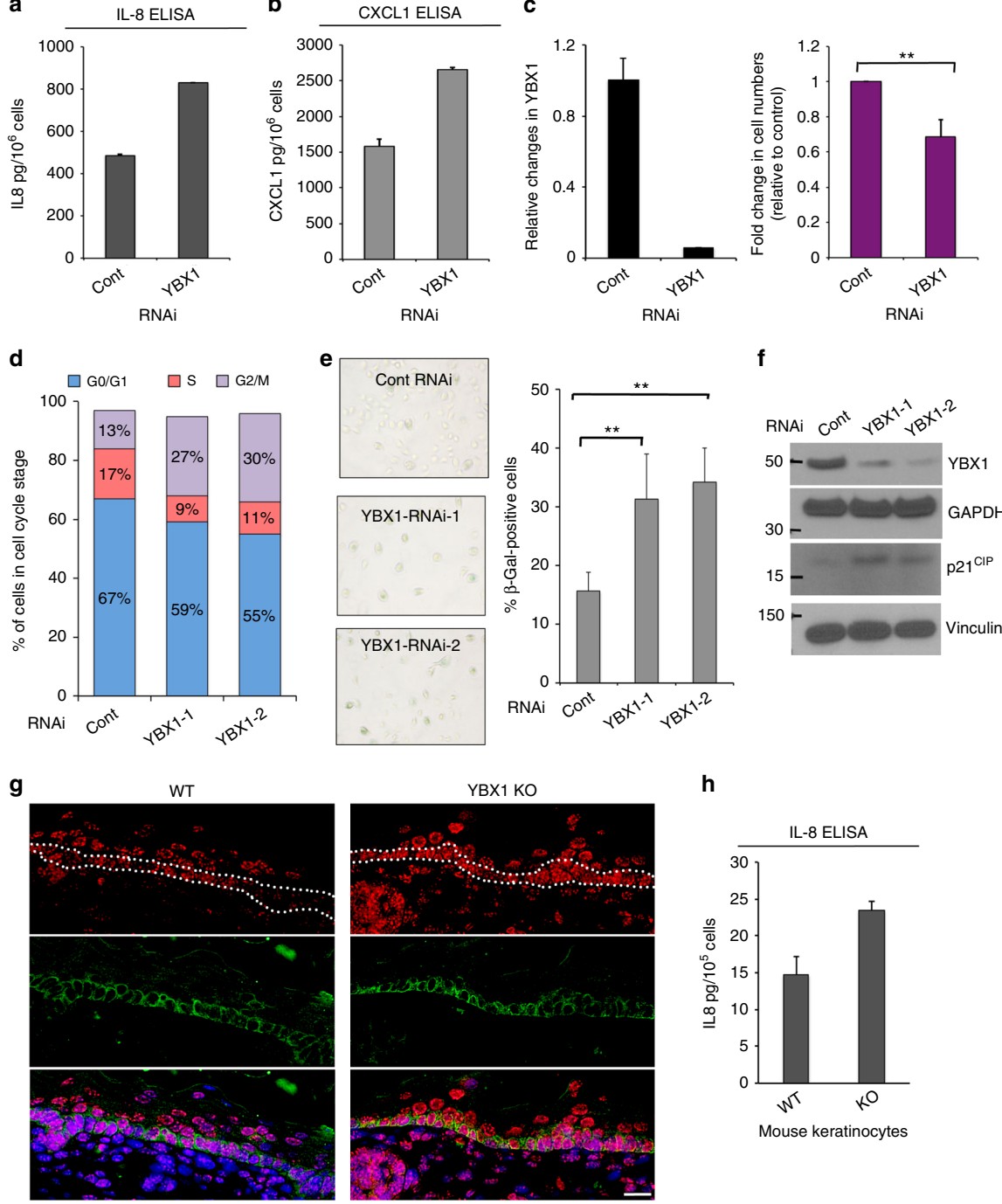

H3K9me3 KRT15 DAPI

whereas such foci are found typically in differentiating cells from the upper epidermal layers (Fig. 7g). Likewise, primary mouse keratinocytes isolated from YBX1 KO mice secreted more mIL-8 when compared to their WT controls (Fig. 7h).

Next we tested the ability of YBX1 controlled cytokines to modulate keratinocytes senescence and epidermal progenitor function in a paracrine manner. Conditioned medium collected from cells with siRNA-mediated depletion of YBX1 prompted significantly (as determined by unpaired $t$-test) higher numbers of primary human keratinocytes to undergo cellular senescence and exhibit positive SA-β-Gal staining as compared to cells grown in control medium (Fig. 8a). The magnitude of increase in SA-β-Gal-positive cells was similar to the increase of senescence in 'aged' keratinocytes (derived from older donors) or in 'young' cells, grown in medium supplemented with factors secreted from 'aged' cultures (Supplementary Fig. 6A). Similarly YBX1 KD-conditioned medium diminished the ability of epidermal progenitors to self-renew and form colonies in feeder layers (Fig. 8b). CXCL1 and IL-8 have been shown to act through the closely related CXC receptors CXCR1 and CXCR2 with CXCR2 being the dominant transmitter of downstream effects as it binds to both chemokines while CXCR1 is selective for IL-8 only[49]. Thus, the biological relevance of YBX1 driven cytokine production for the induction of senescence in human keratinocytes was assessed by a pharmacological blockade of the receptor for CXCL1 and IL-8, CXCR2. Treatment with a highly selective small molecule CXCR2 antagonist (SB 225002, IC50 = 22 nM) was able to partially but significantly (as determined by unpaired $t$-test) rescue the effect of YBX1 depletion for the development of senescence phenotype in primary human keratinocytes as assessed by numbers of SA-β-Gal-positive cells (Fig. 8c). This effect was comparable to the outcome of CXCR2 inhibition for 'young' epidermal progenitors grown in 'aged' keratinocytes medium (Supplementary Fig. 6B). The specificity and efficacy of the pharmacological blockade of cytokine signaling for rescuing the effects of YBX1 was further confirmed by blocking antibodies against both CXCR1 and CXCR2. In agreement with our previous findings, siRNA-mediated depletion of YBX1 in human epidermal progenitors diminished their self-renewal capacity and resulted in the formation of considerably less colonies on feeder layers. This effect was fully rescued when the co-cultures were treated with CXCR1/2 blocking antibodies (Fig. 8d). Similar findings were obtained when the colony-forming assay was performed in the presence of either CXCR2 antagonist or vehicle control (Supplementary Fig. 6C).

Taken together, our findings indicate that the ability of YBX1 to bind to a subset of cytokine transcripts and prevent their translation protects proliferating epidermal progenitors from undergoing replicative senescence.

## Discussion

We identified a specific subset of RBPs expressed in keratinocytes and specifically enriched in the epidermal progenitor population. Among those, YBX1 exhibits a unique ability to control epidermal progenitor cell function through translational inhibition of cytokine biosynthesis and modulation of the cellular senescence program. The requirement for YBX1 in maintaining the epidermal progenitor population has been demonstrated both in vitro, in primary human keratinocytes, and in vivo, using a knockout mouse model. Taken together our data support a model where a loss of YBX1 in the epidermis during development or adult epidermal renewal leads to progenitor cell dysfunction and inappropriate senescence associated with increased production of specific cytokines.

The skin is the largest mammalian organ and functions to provide protection from protean and numerous external insults[50]. A population of cycling progenitor cells in the basal layer of the epidermis regenerates the epidermis every 6 weeks in humans[51–54]. Several distinct populations of stem cells in the murine and human epidermis have been identified that differ in their respective commitment to various lineages under steady-state conditions[53,55,56]. Recent lineage-tracing studies have determined that in order to maintain homeostasis of the inter-follicular epidermis, upon division keratinocyte progenitors choose stochastically between symmetric and asymmetric fate and as a result on average one cell remains in the proliferating compartment, whereas the other commits to differentiation[57–59]. A detailed molecular characterization of the cycling epidermal cell state has begun to emerge in recent years. However, attention to transcriptional regulation and chromatin remodeling has overshadowed efforts to characterize the equally vital role of translational control by RBPs in the regulation of epidermal progenitor function. In this study we began by defining the subset of classical RBPs present in primary cultures of epidermal progenitors through mRNA interactome analysis. Focusing on proteins previously recognized as containing an RNA-binding domain, we analyzed publically available transcriptome data[36] and showed that a substantial number of these RBPs undergo changes in expression level during keratinocyte commitment to differentiation, suggesting a specific role of RNA-binding activity in epidermal progenitor function. This observation was further supported by our detailed analysis of the function of one of these candidate regulators, the cold-shock domain containing RBP YBX1. Notably, another member of the same family of proteins, Lin28, has been implicated in governing stem cell function in other systems and also regulating pluripotency during iPSC generation[10, 60]. While Lin28 is expressed only during embryonic development in the skin, its reactivation in the adult epidermis exerts a profound effect on tissue regeneration[10]. Together with

**Fig. 7** YBX1 depletion promotes cytokine secretion and senescence phenotype. **a**, **b** Protein levels of human IL-8 (**a**) and CXCL-1 (**b**) were analyzed in cellular supernatants from cultures of control or YBX1 siRNA transfected cells using ELISA. After harvesting, the supernatant cell numbers were counted and used for normalization of detected absorbance at 450 nm. Error bars represent mean SD, $n = 3$. **c** YBX1 knockdown in cultures after 4 days of incubation with control or YBX1 siRNA: YBX1 levels were analyzed by qRT-PCR and normalized using 36B4 as a control. Error bars represent mean SD, $n = 3$. Cell numbers in the cultures used in **a**, **b** were calculated and fold change in YBX1-depleted cells was analyzed relative to controls. Error bars represent mean SD, $n = 3$, **$p < 0.01$, unpaired $t$-test. **d** Human keratinocytes were transfected with control or YBX1 siRNAs. Cell cycle distribution was analyzed by flow cytometry using propidium iodide. **e** Human keratinocytes were stained with SA-β-Gal 4 days after transfection with control or two different YBX1 siRNAs. Total and SA-β-Gal-positive cells were counted under light microscopy (×40). Result shows the percentage of SA-β-Gal-positive cells in control and YBX1-depleted cells (**$p < 0.01$, unpaired $t$-test). Error bars represent mean SD, $n = 5$. **f** Protein levels of YBX1 and the senescence marker p21$^{CIP}$ were analyzed by western blotting in control and YBX1-depleted cells using specific antibodies. GAPDH and vinculin were used as loading controls after the same cellular lysates were run on separate SDS-PAGE gels. **g** Analysis of senescence-associated heterochromatin foci in the epidermis of wild type (WT) and YBX1 knockout embryos (YBX1KO) harvested at E.16 using H3K9me3 immunofluorescence staining. DAPI was used to stain nuclei, and KRT15 is a marker for basal keratinocytes. Single channels are shown for better resolution. The basal epidermal layer, where the senescence phenotype is most prominent, is marked with two dotted lines in the red (H3K9m3) channel. Shown images are representative of $n = 4$. Scale bar is 16 μm. **h** Mouse primary keratinocytes were isolated from WT and KO embryos at E.16, placed in culture, and 4 days later conditioned medium was harvested and analyzed for levels of secreted mIL-8 by ELISA. Error bars represent mean SD, $n = 3$

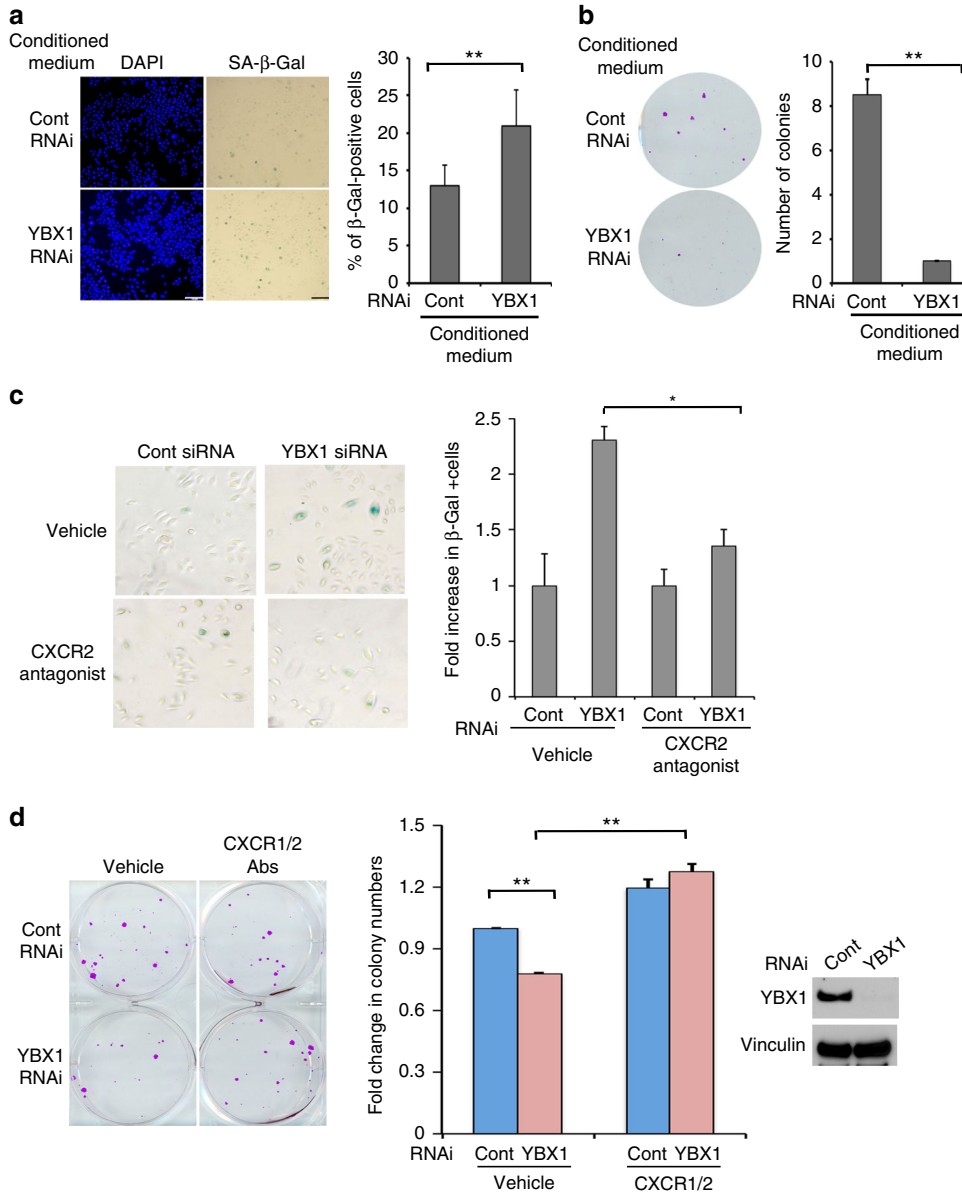

**Fig. 8** The senescence phenotype promoted by YBX1 depletion in epidermal progenitors is dependent on the secretion of senescence-associated CXC cytokines. **a, b** YBX1-mediated induction of paracrine senescence. **a** Conditioned medium from human primary keratinocytes transfected with control and YBX1 siRNA (collected 4 days after transfection) was added to normal proliferating epidermal progenitors. Four days later, the cultures were stained with SA-β-Gal. Total and SA-β-Gal-positive cells were counted under light microscopy. Error bars represent mean SD, $n = 7$, *$p < 0.05$, unpaired $t$-test. **b** A portion from the same conditioned medium from control and YBX KD cells (as in **a**) was added to epidermal progenitors, seeded at clonal density (100 cells per well) on the top of feeder layer from mitomycin C-treated Swiss3T3 fibroblasts. Ten days later, the samples were fixed, the colonies were stained with SRB, and counted using ImageJ, **$p < 0.01$, unpaired $t$-test. **c** CXCR2 small-molecule inhibitor reverts keratinocytes senescence caused by YBX1 depletion. Primary cultures of human keratinocytes transfected either with control or YBX1 siRNA were treated with vehicle or the small-molecule antagonist of CXCR2 and stained for SA-β-Gal. SA-β-Gal-positive cells were counted under light microscopy (×40) and shown as percentage of total cells (*$p = 0.012$, unpaired $t$-test). **d** Blockade of the CXCL1/IL-8 signaling at the level of receptor activation rescues the effect of YBX1 knockdown on epidermal progenitor renewal: primary human keratinocytes transfected with control and YBX1 siRNA were seeded at clonal density (100 cells per well) over Swiss3T3 feeder layer and treated with vehicle control or a mix of blocking antibodies (3 μg each) for the two receptors interacting with CXCL1 and IL-8, CXCR1 and CXCR2. Ten days later, the cultures were fixed, the colonies were stained with SRB, counted using ImageJ, and fold changes in the numbers relative to the control are presented as average values, *$p < 0.05$, **$p < 0.01$, unpaired $t$-test

our current findings on the in vitro and in vivo role of YBX1 in the epidermis, substantial evidence exists for a key regulatory program of keratinocyte stem cell function driven by various RBPs.

Numerous studies have implicated YBX1 in cellular functions such as protein translation, mRNA localization and stability,

transcriptional control, and cell cycle modulation[13,42,61–63]. Deletion of YBX1 in vivo is embryonically lethal due to growth retardation and deficiencies of neural development. Interestingly, the molecular mechanisms underlying these profound effects of YBX1 depletion involve control of stress response pathways including the ability of YBX1 to protect cells from

senescence[38,64]. These data are in line with our observations defining YBX1 as a key suppressor of senescence-associated cytokine translation. Our results show that YBX1 post-transcriptionally regulates cellular senescence by binding to the 3′ UTR of specific senescence-associated cytokine mRNAs. It is also known that the ability of YBX1 to modulate translation is dependent on posttranslational modifications of the protein as well as interactions with other partner RBPs[19,61]. In addition, YBX1 is not only an RBP, it is also a well-characterized transcription factor with ability to translocate from the cytosol to the nucleus. While our data indicate the preferential localization of YBX1 to the cytoplasm of keratinocytes, we cannot exclude the possibility of YBX1 shuttling between the cytosol and the nucleus and binding to either to DNA or to mRNAs. It seems likely that both of these activities contribute to the role of YBX1 in controlling cytokine expression.

In our study, we introduce the role of YBX1-mediated cytokine-driven senescence suppression in proliferating keratinocytes to the many facets of epidermal stem cell functions. Alterations in this process could produce several different clinical scenarios in vivo. Age-mediated decline of YBX1 expression in the skin (Fig. 2) could explain the increasing presence of senescent cells in aging skin and senile defects in tissue renewal (e.g., hair loss, slower wound healing, etc.). In addition, others have shown that age-associated alterations in epidermal stem cell homeostasis are linked to an imbalance of cytokine signaling, which in turn might be part of a broader tumor-suppressive program[4].

Beyond the importance of YBX1 in controlling senescence, the global polysome analysis of primary keratinocyte cultures with or without depleted expression of YBX1 identified additional YBX1-related functions. For example, YBX1 promotes the translation of a subset of proteins mediating EMT such as SHIP1, GPR124, and AGR2. These data support previously published studies describing YBX1 as a regulator of EMT in cancer cells[13]. How these targets are involved in regulating epidermal stem cell behavior is currently under investigation. In summary, we identified a regulatory network that includes the ability of the RBP YBX1 to limit the production of senescence-associated cytokines at translational level in keratinocytes and thus maintain epidermal progenitor function and tissue homeostasis in the skin.

## Methods

**Cell culture**. Human epidermal keratinocytes (HKC) were purchased from ATCC (Virginia, US) or isolated from human skin tissue samples. Primary cultures from donors were isolated following the protocol developed by CELLnTEC Advanced Cell Systems AG and described in details here: http://cellntec.com/products/resources/protocols/isolation/ (CELLnTEC Advanced Cell Systems AG, Bern, Switzerland). Briefly tissue samples were incubated overnight with 5 U dispase solution (prepared from CnT-DNP-10) to detach the dermis from the epidermis. Keratinocytes were further separated using TrypLE™Select (Thermo Fisher Scientific). Primary human keratinocytes were cultured in CnT57 supplemented with IsoBoost for at least the first 3 days post seeding, then cultures were switched to either standard CnT57 medium or to Keratinocyte Serum-free medium (K-SFM, Invitrogen) containing supplements (according to manufacturer's instructions) and 1% antibiotic–antimycotic reagent (Gibco). Cultures were maintained at 37 °C in 5% $CO_2$. Cells were passaged around 60% and split 1:3 by using TrypLE™Select (Thermo Scientific). HKC cultures were used between passages 1 and 5. While there are no identifiers on the selected donors, samples are tested negative for HIV-1, HIV-2, HTLV-1, HTLV-2, Hepatitis B, and Hepatitis C. Due to the high risk work with primary human samples, mycoplasma content is biweekly tested in the laboratory. Cells testing positive or mycoplasma were eliminated.

Mouse primary keratinocytes were isolated from the back skin of mouse embryos (E.16, both female and male animals, strain C57BL/6J) treated with 1 U of dispase solution (StemCell Tech). After overnight incubation at 4 °C, the epidermis was separated from the dermis and placed on TrypLE™ solution (Thermo Fisher Scientific) for the gentle isolation of keratinocytes; 20 min later, CnT07 medium was added to wash the cells and after that the cellular suspension was seeded onto collagen-coated plates.

MEFs were harvested following a standard protocol[65]. Briefly, mouse embryos (both female and male animals, strain C57BL/6J) at E.13–14 were harvested, minced, and trypsin digested. Genomic DNA was isolated for genotyping. After

10 min incubation with the Trypsin solution at 37 °C, MEFs medium (DMEM with 20% FBS) was added, the suspension was centrifuged, cellular pellets were re-suspended in fresh MEFs medium, and plated on 100 mm tissue culture plates. Cells from passages 2–7 were used for experiments.

**Human tissue samples for isolation of primary keratinocytes cultures**. The primary adult keratinocytes were isolated from human skin tissue samples obtained during surgical procedures at Massachusetts General Hospital, Boston, MA, with IRB approval for discarded human tissue (2013P002158/MGH and MGH 2008P001742/2). The IRB at Massachusetts General Hospital/Partners Healthcare has determined that the research activities described in this paper do not meet the definition of human subjects research for the following reasons:

- There will be no intervention of interaction with a living person that would not be occurring or would be occurring in some other fashion, but for this research;
- There will be no identifiable private data/information obtained for this research in a form associable with the individual from whom the human material was obtained. Associable means that the identity of the subject is or may readily be associated with information through direct or indirect identifiers, e.g., codes.

**Mice**. Animal care and experiments were performed in accordance with the Guide for the Care and Use of Laboratory Animal of the National Institutes of Health and were approved by the IACUC at Massachusetts General Hospital. The YBX1 knockout mice were kindly provided by Dr. T. Ley (Washington University School of Medicine, Saint Louis, MO, USA)[38]. The mice were generated and maintained on C57BL/6 background; all histological and immunofluorescence analyses of adult mice tissues were performed using female mice. Genotyping was performed as previously described[38] using the following primers:
PGK1 (5′-TGAGACGTGCTACTTCCATTT)
YBX1-R2 (5′-AGCGGGTCACATTCTTACATAG)
YBX1-F3 (5′-AGGAACGGATAGGTTTCATCA).

**siRNAs and plasmid DNA and methods for transfection**. Predesigned siRNAs for human YBX1 and control siRNAs were purchased from Invitrogen, Carlsbad, CA (Cat. No: s9732, s9733, and 4390843). For the rescue experiment with adenovirus encoding YBX1, a custom made YBX1 RNAi was designed and used (Dharmacon, 5′-CAGTTCAAGGCAGTAAATATGCA-3′). The RNAi-resistant YBX1 adenovirus was constructed by generating point mutations in the parent pCDNA-YBX1 construct using the Quick change site-directed mutagenesis kit (Stratagene California) and the following primers:
YBX1siResistant_FWD 5′GGTGTTCCAGTTCAAGGAAGTAAGTATGCAGC AGACCG
YBX1siResistant_REV 5′-CGGTCTGCTGCATACTTACTTCCTTGAACTGG AACACC.

The adenovirus YBX1(pAdCMV_YBX1siR) was made using the Gateway recombination system with the mutated YBX1 sequence inserted into the pENTR4 vector (Invitrogen) and the pAd/CMV/DEST vector (Invitrogen). For HKC, $5 \times 10^5$ cells were transfected with YBX1 or control siRNA at 50 pmol of each siRNAs by electroporation using Amaxa™ Nucleofector™ Technology and human keratinocyte nucleofector kit ((Lonza AG) according to manufacturer's direction. The 3′UTR luciferase reporter constructs (for 3′-UTR_CXCL1, 3′-UTR_IL8, and control vector) were purchased from ABM, Inc (MT-h11050, MT-h05305, and m012). Deletion series using the 3′-UTR_IL8 plasmid were generated according to the identified putative AREs (utilizing restriction enzymes PstI, BamHI, KpnI, and XhoI). YBX1 WT and mutant plasmid containing amino acids 128–324 were previously described[44].

Plasmid DNAs were transfected in MEFs using Lipofectamine 3000 (Invitrogen).

**Validation of RNA-binding ability by PNK assay**. Primary human keratinocytes were lysed in buffer containing 100 mM NaCl, 50 mM Tris-HCl, pH 7.5, 1 mM $MgCl_2$, 0.1 mM $CaCl_2$, 0.1% SDS, and 0.5% Na-deoxycholate; 500 µg of each protein lysate was incubated with 12.5 U of DNAse I and also with serial dilutions of RNAse I (1000 U, 100 U, 10 U) at 37 °C for 15 min; 10 µl of protein A/G magnetic beads (Millipore, LSKMAGAG02), pre-coated with 5 µg of YBX1 antibody or control mouse IgG, were added to each lysate and incubated for 2 h at 4 °C. Beads were washed with lysis buffer and labeled with [γ-32P]ATP (PerkinElmer) prior incubation with T4 protein kinase. The labeled beads were washed with PNK buffer (50 mM NaCl, 50 mM Tris-HCl, pH 7.5, 10 mM $MgCl_2$, 0.5% SDS), and separated on SDS-PAGE. Transferred membranes were visualized by auto-radiography. The confirmatory YBX1 western blot was carried out using YBX1 antibody (Epitomics-Abcam, 2397-1, 1:10,000).

**Colony forming assays**. Primary HKC were transfected with control or YBX1-siRNAs as described above and seeded at the indicated cellular density in 6-well plates. In the case of feeder layer co-cultures (when adult keratinocytes were used),

Swiss3T3 cells were treated with mitomycin C and plated in culture plates 24 h prior to seeding of the transfected keratinocytes. Clonal density cultures were maintained either in K-SFM medium or in FAD medium (50% DMEM, 50% DMEM/F12 with the addition of 12.5% FBS, 5 μg/ml insulin, $1.8 \times 10^{-4}$ M adenine, 0.5 μg/ml hydrocortisone, $1 \times 10^{-10}$ M cholera toxin and 10 ng/ml EGF) for 10 days, fixed with 50% of trichloroacetic acid (TCA), and stained with sulforhodamine B (SRB, G-Biosciences)

**Flow cytometry and cell cycle.** Primary HKC were grown until 70% confluence, washed with PBS and harvested; $1 \times 10^5$ cells per condition were incubated for 30 min on ice with the following antibodies: APC-conjugated anti CD71 (BioLegend, 334108) and Pacific Blue-conjugated anti ITGA6 (BioLegend, 313620). FACS analysis was performed using LSRII FACS Analysers (BD Biosciences) and analyzed by the Flowjo software at the Ragon Institute (Cambridge, MA). The cell cycle was detected by using propidium iodide (PI) (Invitrogen). Cells were fixed in 70% ethanol and resuspended in 10 μg/ml of PI and 50 μg/ml of RNaseA containing PBS solution. Cells were analyzed using the BD facsCANTO flow cytometer (BD Biosciences) and FlowJo software.

**RNA isolation and qRT-PCR.** For tissues: C57BL/6 mouse embryos were collected at E.14–18 and total RNA was purified from the back skin tissue using Trizol (Invtirogen). For cells: total RNA from cells was isolated using RNAeasy microkit (Qiagen). Equal amounts of RNA were reverse-transcribed using Maxima First strand cDNA synthesis kit (Thermo Scientific). qRT-PCR was performed with Kapa SYBR qPCR master mix (Kapa Biosystems) and gene-specific primers using Bio-rad iCycler instrument. Relative levels of expression were determined using the Ct method relative to the housekeeping genes, 36B4 and YWHAZ. The specific primers for qRT-PCR are listed in Supplementary Table 1.

**Western blotting.** For western blotting, following primary antibodies were used: anti-YBX1 (Epitomics-Abcam, 2397-1, 1:10,000), anti-PTBP (Cell signaling, 8776s, 1:1000), anti-Histone H3 (Santa Cruz, sc-10809, 1:1000), anti-KRT1 (Covance-BioLegend, PRB-149, 1:1000), anti-Vinculin (Santa Cruz, sc-73614, 1:1000), anti-p21 (Cell signaling, 2947s, 1:1000), anti-GAPDH (Origene, TA802519, 1:2000). Donkey anti-rabbit IgG-HRP (Santa Cruz, sc-2313) and donkey anti-mouse IgG-HRP (Santa Cruz, sc-2318) secondary antibodies were used at 1:5000. Cells were lysed in RIPA buffer and total protein amounts were normalized by BCA protein assay kit (Thermo Scientific). SDS-PAGE was performed using pre-cast NuPAGE Bis-Tris 4–12% mini-gels (Invitrogen) with MES buffer, following the manufacturer's instructions. Silver staining was performed using Silver Quest staining kit (Invitrogen). All uncropped western blots can be found in Supplementary Fig. 7.

**[35S]-methionine incorporation.** Primary HKC were transfected by electroporation with control or YBX1 siRNAs and seeded into 6-well plates. After 4 days, cells were incubated with Met-Cys-free medium (US biological) for 15 min and then switched to 0.1 mCi/ml of [35S]-methionine (Perkin-Elmer, NEG009A500UC) containing medium. Following 1 h of incubation, the cells were washed and harvested in RIPA buffer. [35S]-methionine incorporation into proteins was detected after TCA precipitation by SDS-PAGE. For TCA precipitation, 10 μl of cellular lysate was spotted on 3 mm filter paper, air dried and placed in ice-cold 10% TCA solution. After 20 min, filter papers were washed twice with 10% TCA and ethanol, respectively, and air dried for 30 min. Isotope activity was detected using Scintillation counter (Beckman, LS6000IC) and normalized to protein amount.

**Histology and immunofluorescence.** Mouse skin from the back of 16-day-old embryos was harvested and fixed in 10% formalin or 4% paraformaldehyde overnight for paraffin or frozen sections, respectively. Paraffin sections (immobilized on slides and dewaxed) were used for Hematoxylin and Eosin (H&E) staining and histological analysis. H&E staining was carried out following the standard protocol (http://www.ihcworkd.com). Slides were mounted in Entellan New rapid mounting media (Electron Microscopy Sciences). Frozen sections (mounted in OCT embedding compound and frozen at −80 °C) were used for immuno-fluorescence staining: primary antibodies were incubated for 3 h, and secondary antibodies were incubated for 1 h at room temperature in 5% BSA/PBST. Nuclei were stained with DAPI (Invitrogen), and the slides were mounted in Prolong Gold Antifade Mount (Invitrogen). Following primary antibodies were used: anti-KRT5 (ProGen Biotechnik GmbH, GP-CK5, 1:500), anti-KRT15 (Covance-BioLegend, PCK-153P, 1:200), anti-Loricrin (Covance-BioLegend, PRB145P, 1:200), anti-KRT1 (Covance-BioLegend, PRB-149, 1:200), anti- Ki67 (Abcam, ab16667, 1:200), anti-YBX1 (Epitomics–Abcam, 2397-1, 1:500), anti-H3K9me3 (Epigenteck, A4036-025, 1:200). Alexa-conjugated secondary antibodies, Alexa Fluor 488 and 568 (Invitrogen, A11039, A11073 and A10042, 1:1000), and Alexa 488 phalloidin (Invitrogen, A12379, 1:1000) were used. Pictures were acquired with a FSX100 microscope (Olympus).

**Polysomal RNA purification.** Primary HKC cultures pooled from different donors were transfected with control or YBX1 siRNAs by electroporation. Four days later

(to achieve a maximum suppression of YBX1 expression as analyzed by western blot), the cells were placed in 50 μg/ml of Cycloheximide (Sigma) containing medium for 5 min, and lysed in RSB buffer (10 mM Tris, pH 7.4, 10 mM NaCl, 15 mM MgCl$_2$) containing 1.2% Triton X-100, 1 mM DTT, 50 μg/ml Cycloheximide, and 3% deoxycholate. The lysates were centrifuged for 5 min at 4 ° C to pellet the nuclei. Supernatants were layered over 10–45% sucrose gradients and centrifuged at 37,000 rpm for 2 h in an SW41 (Beckman) rotor at 4 ℃. The gradients were fractionated using Density Gradient Fractionation System (Brandel, BR-188). Two biological replicates of polysomal RNAs and total RNAs were isolated using Trizol reagent (Invitrogen) and cleaned up with RNeasy kit (Qiagen) before subsequent qRT-PCR or high-resolution RNA sequencing (RNA-seq) analysis.

**RNA-seq and bioinformatics analysis.** RNA-seq and bioinformatics analysis were performed at the Genomics and Proteomics Center of the Beth Israel Deaconess Medical Center (BIDMC). RNA-seq was carried out at a depth of 15–20 million single-end reads. Briefly, libraries for sequencing were generated from the double-stranded cDNA using the Illumina TruSeq kit according to the manufacturer's protocol. Library quality control was checked using the Agilent DNA High Sensitivity Chip. Sequencing was performed on an Illumina HiSeq 2000. The differentially expressed genes were identified on the basis of multiple-test corrected p-value and fold change. Genes were considered significantly differentially expressed if the p-value was <0.05 and absolute fold change >2.

**RNA immunoprecipitation.** Primary HKC were harvested in modified RSB buffer (10 mM Tris, pH 7.4, 10 mM NaCl, 15 mM MgCl$_2$, and 1% deoxycholate). RNA-IP was performed using a Magna RIP$^{TM}$ RBP immunoprecipitation kit (Millipore) following the manufacturer's instruction. Briefly, pre-cleared (overnight) cell lysates were incubated with 5 μg of YBX1 antibody or rabbit IgG (RN015P, MBL Life Science, Japan), respectively. Immune complexes were recovered by Protein A/ G magnetic beads. RNAs were extracted using RNAeasy kit (Qiagen) with DNAse treatment, and mRNA levels were measured by qRT-PCR using the primers listed below. Binding to the target RNA was calculated as a percent of the input using fold enrichment methods. YBX1 binding to target mRNAs is presented as a fold increase compared to non-specific IgG. The specific primers used for RNA-IP are listed in Supplementary Table 1.

**In vivo isolation of human keratinocyte RBPs.** mRNA interactome capturing experiments and mass spectrometry were carried out as previously described[33–35]. Briefly, HKC were grown on $100 \times 10$ cm dishes and irradiated with 0.15 J/cm$^2$ UV light at 254 nm. Cells were immediately harvested in ice-cold lysis buffer (20 mM Tris, (pH 7.5), 500 mM LiCl, 0.5% lithium dodecyl sulfate, 1 mM EDTA, 5 mM DTT) and homogenized using a 27G needle. Poly(A+)-tagged mRNAs and cross-linked polypeptides were captured with oligo(dT)25 magnetic beads (NE Biolabs). Luciferase spike-in control RNA (Promega, L4561) was added during the capture and used for normalization of qRT-PCR results. Eluted and purified proteins were processed following the standard protocol as previously described[33–35]. Three biological replicates were prepared: for each replicate, pooled cultures from different donors were used. The specific primers used in the mRNA interactome capturing experiments are listed in Supplementary Table 1.

**ELISA and SA-β-gal staining and treatment with the CXCR2 inhibitor and the CXCR antibodies.** Primary HKC were transfected with control or YBX1 siRNAs by electroporation, and 4 days after transfection the culture supernatants were harvested for ELISA. IL-8 and CXCL-1 ELISA kits were purchased from R&D systems and ELISA-based detection was carried out following manufacturer's directions. The concentration of the cytokines in the culture supernatant (pg/ml) was calculated and normalized to respective cell numbers. The absorbance at A450 nm was measured using a microplate reader. Cellular senescence was detected using an SA-β-gal-based senescence detection kit (Abcam). Cells were observed under a light microscope with 100× total magnification and the images were taken using Canon EOS digital camera. Total versus blue-green SA-β-gal-positive cells were counted in different areas of the image. DMSO (0.1%) or 50 nM of CXCR2 inhibitor SB225002 (Tocris) or a mixture of CXCR1/2 blocking antibodies (3 μg each: anti-hCXCR1, MAB330, and anti-hCXCR2, MAB331, R&D system) were used to treat HKC cultures prior to SA-β-gal staining or during the colony-forming assay.

**Data availability.** The authors declare that all data supporting the findings of this study are available within the article and its Supplementary information files or from the corresponding author upon reasonable request.

RNA sequencing data have been deposited in the NCBI Gene Expression Omnibus (GEO) database under the accession code: GSE110014.

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

## Acknowledgements

We thank Dr. Tim Ley's group for providing the YBX1 knockout mice, Dr. Donald Bloch's group for providing the YBX1 mutant plasmid, the EMBL Proteomics Core Facility for expert mass spectrometric analyses of the epidermal progenitor interactome, and the Genomics, Proteomics Bioinformatics and Systems Biology Center of Beth Israel Deaconess Medical Center for the RNA-seq analysis. This work is partially supported by a seed grant of the Harvard Stem Cell Institute (Boston, USA).

## Author contributions

E.K. performed the majority of the experiments. K.T. and J.W. processed the human tissue samples and primary cell cultures and assisted with the experiments. R.H. performed and analyzed the mRNA capture study. K.K.L. and G.L.N. assisted with the bioinformatics analysis. V.A.N., P.H.S., S.W.L., and M.W.H. provided critical scientific support and conceptual insight for the project. A.M. conceived the study, led the overall analysis, and wrote the manuscript with E.K.

## Additional information

**Competing interests:** The authors declare the following competing interests: The experiments described in this paper were partially funded through a collaborative agreement between Shiseido Inc. (Japan) and the Cutaneous Biology Research Center at Massachusetts General Hospital, Harvard Medical School. S.W.L., M.W.H., and A.M. are inventors on a patent application entitled "Methods and compositions for promotion of skin rejuvenation and healing" covering the utility of YBX1 modulation for affecting skin homeostasis and filed by Massachusetts General Hospital.

