## [Peer Review File · Nature Communications]

Reviewers' Comments:

Reviewer #1 (Remarks to the Author)

In the manuscript, "Maintenance of epidermal progenitors by YBX1 mediated post-transcriptional control" Kwon. Et al. identified YBX1 as an RNA binding protein that is selectively expressed in epidermal progenitor populations. Knockout of YBX1 in mice resulted in decreased hair follicle formation and alterations in the interfollicular epidermis. In human keratinocytes, knockdown of YBX1 resulted in decreased proliferation and an increase in senescence. Interestingly, the authors find that many of the cytokines associated with senescence is increased on a translational level upon YBX1 knockdown. The authors suggest that YBX1 controls epidermal progenitor function through the negative translational regulation of senescence associated cytokines. While this manuscript is potentially interesting, it suffers from several major concerns which are described below:

A) One of the authors' main points in the paper is that YBX1 promotes epidermal progenitor function, however the phenotype in YBX1 knockout skin is weak. There is no changes in the expression of differentiation proteins in the epidermis nor is their any major changes to the basal layer of the epidermis where the stem and progenitor cells resides. The only noticeable changes are that there are fewer ki67 positive cells reflecting that there may be differences in the proliferation rate of the basal layer cells (This should be quantitated). Since the knockout mice die shortly afterbirth, the authors should take skin from YBX1 knockout mice and control mice and graft it onto NUDE mice to see the long-term impacts of YBX1 knockout on the skin. If YBX1 is truly necessary for progenitor cell function, the knockout skin should atrophy over time and result in a loss of the basal layer. This will also allow the authors to determine the long-term impacts of YBX1 loss on hair follicle maintenance (does the grafted skin lose hair over time).

B) Another of the authors' main points is that YBX1 negatively regulates cytokine translation to prevent premature senescence. However the in-vivo data supporting this is lacking. The evidence provided in-vivo is staining for H3K9me3 showing that there are higher levels of H3K9me3 in YBX1 knockout skin. This data is hard to interpret and should be quantitated. The images should be separated and not merged into one figure since it is hard to validate the H3K9me3 intensity when it is merged with the DAPI. There also needs to be other orthogonal assays shown that validate the in-vivo senescence phenotype. The authors should do polysomal fractionation assays on WT and KO mouse epidermis and determine if there is higher levels of translation of the senescence associated cytokines. The authors should also do Western blot analysis to determine if there are higher levels of P16 and P21 in YBX1 KO mouse epidermal tissue as compared to WT.

C) The authors suggest that YBX1 targets the 3'UTR of cytokines to repress their translation. This was demonstrated using the 3'UTR of IL-8 fused to a Luciferase reporter. The results from this are unclear. First, the experiments were performed in 293T cells rather than Human keratinocytes. 293T cells should not be used since they are already transformed and are different than primary human keratinocytes. Second, it is unclear if the IL-8 reporters are actually responsive to YBX1 levels since there is already a 2 fold change in Luciferase activity in YBX1 knockdown cells in just the vectors alone while there is only a 3 fold change upon YBX1 knockdown in IL-8 reporters. Lastly and most importantly, the authors need to identify the sequence motif or secondary structure that YBX1 is binding to in the cytokine mRNAs and validate the importance of the sequence using reporter assays.

D) To definitively demonstrate that YBX1 loss leads to increased senescence due to increased translation of cytokines the authors should do the following experiments:

1. Since this should be a non-cell autonomous mechanism, the authors should incubate wildtype cells with the conditioned medium of YBX1 knockdown cells (does this lead to increased senescence for the WT cells?) This could also be done with co-culturing the cells.
2. In addition to the CXCR2 antagonist (which may have off target effects), the authors should do a double knockdown of YBX1 with CXCR2 to determine if that reverses the increased senescence

seen in the YBX1 knockdown cells. Does this double knockdown also reverse the loss of proliferation phenotype?

E) Is there a difference in YBX1 expression between "aged" and "young" keratinocytes? Does overexpression of YBX1 reduce the percentage of senescent cells in "aged" keratinocytes?

Minor issues:

1. In fig.2B&C, why was the mRNA level of Ybx1 only tested in suspension differentiation conditions at early stage while the protein level of ybx1 was tested in two differentiation conditions at late stage, these two figures should be consistent.
2. In fig. 2E, the author should use a higher magnification (40x or higher) image to clearly demonstrate the localization of ybx1. The images here are too fuzzy to draw any conclusion. It may also help to separate the images from the 3 different channels.
3. Fig. 4A and 4C requires quantitation and statistics to determine significance.
4. In fig. 6B, the reporter assay would be more relevant if performed in keratinocytes, since transformed cells line may have dysregulated cytokine signal pathways. In the same figure, the mRNA level of these reporters should also be included. What is the control vector used in the figure? Does it contain any nonspecific 3'UTR?
5. Statistical analysis is required for fig. 7A
6. In Fig. 7D, the scale bar is still missing. It will be more convincing if the author could prove the regulation of Ybx1 on various cytokines in vivo by staining for IL8/ CXCL1 or their downstream signals molecules in the WT and ybx1 -/- sections.

Reviewer #2 (Remarks to the Author)

The authors report that YBX-1 (also called YB-1) contributes in maintenance of epidermal progenitors via its posttranslational mechanism.

This study may provide some novel information to the related field. However, the manuscript lacks some required data and several critical experiments as indicated in Major comment. Therefore, I cannot recommend the manuscript for publication in Nature Communication in the present form, because of the high status of the journal.

Major comments

1. YBX-1 has some domains such as alanine-proline rich N-terminal domain, cold shock domain NLS, and C-terminal containing B/A repeats. The authors should determine the functional domains of YB-1 required in the process.
2. To further demonstrate post-transcriptional function of YB-1 in maintenance of epidermal progenitors, the authors should perform some rescue experiments using wild-type and some mutants YB-1 as characterized above. In the knocking-down experiment, siRNA resistant wild-type and mutants YB-1 should be used. Moreover, in the experiment using knockout mouse, YB-1-adenovirus may be applicable to embryonic epidermis.
3. To further demonstrate post-transcriptional function of YB-1 in maintenance of epidermal progenitors, the authors should perform quantitative proteome analysis of the protein expression patterns in the above knocking-down/add-back experiment. Moreover, in the experiment using knockout mouse in treatment with YB-1-adenovirus the authors had better perform the proteome analysis.

Reviewer #3 (Remarks to the Author)

In their manuscript Kwon and colleagues determine the RNA-binding protein YBX1 as essential regulator of epidermal progenitor cell populations. The authors identify YBX1 in an unbiased screen for RNA-binding proteins in primary human keratinocytes that are down-regulated with terminal differentiation. Deletion of YBX1 in human keratinocytes and the mouse epidermis reduces cell

proliferation of progenitor populations but does not affect terminal differentiation. Instead, deletion of YBX1 induces cell cycle arrest and cellular senescence. Mechanistically the authors show that YBX1 suppresses translation of IL-4 and CXCL1 mRNAs, and thereby protects epidermal progenitor cells from cytokine-mediated senescence.

YBX1 is a multifunctional protein and its up-regulation is found in aggressive tumours. In addition, YBX1 has been implicated in regulating cellular senescence before, yet the underlying mechanism how YBX1 opposes senescence are unknown. Therefore, the manuscript is potentially interesting for the field.

Comments:

1. Figure 2B. As a positive control, involucrin (or any other terminal differentiation factor) expression should be shown.
2. The authors are very sparse with explaining how they differentiated the keratinocytes. They should elaborate in more detail how the suspension-induced differentiation assay and the spontaneous differentiation assays have been performed.
3. Figure 3. Even when the mice have been published already, the author need to provide evidence that YBX1 RNA and protein expression is lost in the knockout mouse epidermis. In addition, the consequences of YBX1-deletion on epidermal progenitor proliferation and differentiation need quantification.
4. Since the authors study a total knockout mouse for YBX1, it is not clear from figure 3 whether the phenotype is caused by a direct role of YBX1 in the epidermis or whether the skin phenotype is rather due to the overall developmental delay of the mouse embryo. At the very minimum, the authors should measure colony forming efficiency of the mouse keratinocytes. It would be also interesting to know whether the IL8 and CXCL1 are conserved targets of YBX1 in the mouse.
5. Figure 4. Can the effect on proliferation and cell cycle be rescued by an siRNA-insensitive YBX1-construct or treatment with the CXCR2 antagonist?
6. The authors conclude from figure 4 that 'These findings indicate that while YBX1 maintains epidermal progenitor homeostasis, it does not directly target the transcriptional control of keratinocytes commitment to differentiation'. Measuring the RNA expression of only three differentiation markers (Fig. 4D) does not warrant such a strong statement. It is not clear to me why the authors do not provide further evidence for this claim using the RNA seq data obtained from these cells. Also, the RNA seq data are not provided with the manuscript and no link to a public database is provided.
7. Figure 6B. The luciferase assay is not convincing given that also the control vector seems to significantly increase in luciferase activity. Why are no data shown for CXCL1 and 2 or the positive control YBX1 itself?
8. Figure 7D needs to be confirmed by Western blot for H3K9me3.
9. The final conclusion of the authors is that 'YBX1 is able to bind to a subset of cytokine transcripts and prevent their translation thus protecting proliferating epidermal progenitors from undergoing replicative senescence'. This claim is not really supported by data and too strong given that no rescue experiments on keratinocyte proliferation and cell cycle (see also point 5) are provided.
10. No table S4 was provided but it is mentioned in the text.
11. Please check text and panel labelling for figure 7.

Reviewers' comments:

Reviewer #1 (Remarks to the Author):

In the manuscript, "Maintenance of epidermal progenitors by YBX1 mediated post-transcriptional control" Kwon. Et al. identified YBX1 as an RNA binding protein that is selectively expressed in epidermal progenitor populations. Knockout of YBX1 in mice resulted in decreased hair follicle formation and alterations in the interfollicular epidermis. In human keratinocytes, knockdown of YBX1 resulted in decreased proliferation and an increase in senescence. Interestingly, the authors find that many of the cytokines associated with senescence is increased on a translational level upon YBX1 knockdown. The authors suggest that YBX1 controls epidermal progenitor function through the negative translational regulation of senescence associated cytokines. While this manuscript is potentially interesting, it suffers from several major concerns which are described below:

A) One of the authors' main points in the paper is that YBX1 promotes epidermal progenitor function, however the phenotype in YBX1 knockout skin is weak. There is no changes in the expression of differentiation proteins in the epidermis nor is their any major changes to the basal layer of the epidermis where the stem and progenitor cells resides. The only noticeable changes are that there are fewer ki67 positive cells reflecting that there may be differences in the proliferation rate of the basal layer cells (This should be quantitated). Since the knockout mice die shortly afterbirth, the authors should take skin from YBX1 knockout mice and control mice and graft it onto NUDE mice to see the long-term impacts of YBX1 knockout on the skin. If YBX1 is truly necessary for progenitor cell function, the knockout skin should atrophy over time and result in a loss of the basal layer. This will also allow the authors to determine the long-term impacts of YBX1 loss on hair follicle maintenance (does the grafted skin lose hair over time).

We thank the reviewer for pointing out the incomplete characterization of the YBX1 KO phenotype. We performed a more detailed analysis of the mutant epidermis using multiple KO animals. In the revised Fig. 3A, B we are now showing that YBX1 depletion leads to a decrease in the epidermal thickness as well as to the development of significantly less hair follicles when compared to wild type skin. The additional analyses allowed us also to quantify the Ki67 staining and we are now showing the clear reduction of the proliferative capacity in the mutant epidermis.

We apologize for not pointing out clearly that the mutant animals do not survive to birth (very rare exceptions were published but we were able to collect only one double knockout live newborn mouse in the two years we have been using the model). We were able to harvest sufficient number of embryos at day E. 15-16 and perform the experiments described in the paper but the low yield prevented us to successfully transplant mutant skin on wild type nude mice, although we attempted to do so. Therefore we expanded our studies in the human system and also in primary mouse keratinocytes derived from the YBX1 KO mice and believe that we were bale to present compelling evidence on the progenitor failure caused by YBX1 depletion.

B) Another of the authors' main points is that YBX1 negatively regulates cytokine translation to prevent premature senescence. However the in-vivo data supporting this is lacking. The evidence provided in-vivo is staining for H3K9me3 showing that there are higher levels of H3K9me3 in YBX1 knockout skin. This data is hard to interpret and should be quantitated.

The images should be separated and not merged into one figure since it is hard to validate the H3K9me3 intensity when it is merged with the DAPI. There also needs to be other orthogonal assays shown that validate the in-vivo senescence phenotype. The authors should do polysomal fractionation assays on WT and KO mouse epidermis and determine if there is higher levels of translation of the senescence associated cytokines. The authors should also do Western blot analysis to determine if there are higher levels of P16 and P21 in YBX1 KO mouse epidermal tissue as compared to WT.

We agree with the reviewer and are now presenting single channel images of the H3K9me3 staining, where we marked the basal layer of the epidermis using Keratin 15 staining in order to better define the differences between WT and KO animals (Fig. 7G). In addition, we were able to isolate primary mouse keratinocytes from these mice and measured their ability to secrete the senescence associated IL-8 by ELISA. These findings are now included in the revised Fig. 7H and are showing the increased production of IL-8 by YBX1 KO epidermal progenitors.

As we pointed out, the yield of primary mouse keratinocytes from mutant embryos is very low due to the poor survival rates and therefore we isolated only limited amount of mRNA. We further separated the polysomal from non-polysomal fractions and are now showing that although the translation of housekeeping transcripts such as m36B4 is not affected, the polysome-associated fraction of the CXCL1 mRNA is significantly increased in the YBX1 KO epidermal progenitors (Fig. 5F). These results are in agreement with our findings in the human system pointing to an increase in the mRNA translation of senescence-associated cytokines upon YBX1 depletion.

C) The authors suggest that YBX1 targets the 3'UTR of cytokines to repress their translation. This was demonstrated using the 3'UTR of IL-8 fused to a Luciferase reporter. The results from this are unclear. First, the experiments were performed in 293T cells rather than Human keratinocytes. 293T cells should not be used since they are already transformed and are different than primary human keratinocytes. Second, it is unclear if the IL-8 reporters are actually responsive to YBX1 levels since there is already a 2 fold change in Luciferase activity in YBX1 knockdown cells in just the vectors alone while there is only a 3 fold change upon YBX1 knockdown in IL-8 reporters. Lastly and most importantly, the authors need to identify the sequence motif or secondary structure that YBX1 is binding to in the cytokine mRNAs and validate the importance of the sequence using reporter assays.

We agree with the reviewer that the 293T cells are not the most appropriate system for these reporter assays but at the same time primary cultures of epidermal progenitors are not suitable for transient transfections with cDNA constructs as it will be required to execute the experiments. Therefore we decided to utilize the YBX1 KO mouse model and resorted to the isolation of mouse embryonic fibroblasts (MEF) from WT and mutant mice. We then used this system to perform our reporter assays and are now showing in the revised Fig. 6B that the luciferase signal from the IL-8 3'UTR reporter construct is significantly elevated in the YBX1 KO cells as compared to WT. We did not detect any differences in the luciferase translation from the control reporter containing a minimal 3'UTR (Fig. 6B). We next used the YBX1 KO MEFs and performed reverse experiments by co-expressing full length YBX1 cDNA together with the CXCL1 3'UTR reporter and were able to demonstrate that, as expected, this led to the rescue of the elevated translation of CXCL1 and decrease in the luciferase signal. The difference between transfection with control and YBX1 cDNA is significant and corresponds to the magnitude of change observed between WT and KO MEFs. Furthermore we used in the same system an additional YBX1 cDNA encoding a deletion of the cold shock domain of the

YBX1 protein. The cold shock domain has been implicated previously as essential for the ability of YBX1 to bind to its target transcripts. In agreement with these findings, our results show now that the YBX1 deletion mutant fails to modulate the CXCL1 reporter activity in the KO MEFs (Fig. 6C).

We agree with the reviewer that characterizing in more details the 3'UTR sequences responsible for the observed effect of YBX1 is important for the impact of our study. Therefore we performed an unbiased bioinformatics analysis of the 3'UTRs of all transcripts identified to be under the YBX1 control in the RNA seq analysis of the polysomal mRNA fractions. We found various AU-rich elements (ARE), which were enriched in the 3'UTRs of 13 out of the 23 negatively regulated by YBX transcripts (please note that 3 of those are ncRNA/pseudogenes and don't have annotated UTRs). We are presenting now this analysis in the new SFig. 4. It is well established that modulation of translation via ARE is a prominent mode of regulation of cytokine production by RNA binding proteins. Next we generated IL-8 3'UTR deletion mutants lacking clusters of AREs and performed similar promoter analysis in the YBX1 KO MEFs as described above for Fig. 6B. Our data point to a key role of the two proximal AREs for the YBX1 driven translational modulation of IL-8. These findings are now included in Fig. 6D.

D) To definitively demonstrate that YBX1 loss leads to increased senescence due to increased translation of cytokines the authors should do the following experiments:
1. Since this should be a non-cell autonomous mechanism, the authors should incubate wildtype cells with the conditioned medium of YBX1 knockdown cells (does this lead to increased senescence for the WT cells?) This could also be done with co-culturing the cells.

We considered this suggestion and are now presenting in the new Fig. 8A data showing the ability of growth medium conditioned by control and YBX knockdown (KD) keratinocytes to induce cellular senescence in human epidermal progenitors in a paracrine manner. In addition, we supplemented these experiments with analysis of the self-renewal capacity of the epidermal progenitors and were able to show that the same conditioned medium collected from YBX KD cells is able to significantly reduce the clonogenic ability of human primary keratinocytes (Fig. 8B).

2. In addition to the CXCR2 antagonist (which may have off target effects), the authors should do a double knockdown of YBX1 with CXCR2 to determine if that reverses the increased senescence seen in the YBX1 knockdown cells. Does this double knockdown also reverse the loss of proliferation phenotype?

Small molecule inhibitors such as the CXCR2 antagonists have indeed the potential to cause off-target effects and we agree with the reviewer that orthogonal assays are needed to establish the functional significance and specificity of the cytokine pathway. We attempted to utilize siRNA-mediated knockdown of CXCR2 but due to the need of co-transfection with YBX1 siRNAs we encountered high levels of toxicity in the primary cultures. Therefore we resorted to the use of blocking antibodies for CXCR2, which have proven efficacy and also specificity. We used antibodies for both CXCR1 and CXCR2 due to the fact that IL-8 has been shown to partially signal through both receptors. Our new set of findings now indicates that the blocking antibodies for CXCR1/2 are able to rescue the effects of YBX1 depletion on epidermal progenitor function (Fig. 8D and SFig. 5C).

E) Is there a difference in YBX1 expression between "aged" and "young" keratinocytes? Does overexpression of YBX1 reduce the percentage of senescent cells in "aged" keratinocytes?

Change of YBX1 expression during epidermal aging certainly is an important question for the impact of this pathway on skin biology. However, given the complicated nature of the skin aging process, which combines both chronological and environmentally induced aging and is known to be controlled via different mechanisms depending on the location of the epidermis within the body, we believe that studying this process is outside of the scope of the current manuscript. We intent to explore the role of YBX1 in skin aging in depth and have already initiated this line of work. We are including here our preliminary data indicating indeed that YBX1 is decreased during aging. However, we feel that the number of donors tested and the lack of variety between sun-exposed and non-exposed areas warrant cautious interpretation and we would refrain from including it yet into a publication.

Minor issues:

1. In fig.2B&C, why was the mRNA level of Ybx1 only tested in suspension differentiation conditions at early stage while the protein level of ybx1 was tested in two differentiation conditions at late stage, these two figures should be consistent. **Corrected**
2. In fig. 2E, the author should use a higher magnification (40x or higher) image to clearly demonstrate the localization of ybx1. The images here are too fuzzy to draw any conclusion. It may also help to separate the images from the 3 different channels. **Corrected: additional images are provided.**
3. Fig. 4A and 4C requires quantitation and statistics to determine significance. **Corrected**
4. In fig. 6B, the reporter assay would be more relevant if performed in keratinocytes, since transformed cells line may have dysregulated cytokine signal pathways. In the same figure, the mRNA level of these reporters should also be included. What is the control vector used in the figure? Does it contain any nonspecific 3'UTR? **New data in a different, more relevant experimental system is provided.**
5. Statistical analysis is required for fig. 7A **Provided**
6. In Fig. 7D, the scale bar is still missing. It will be more convincing if the author could prove the regulation of Ybx1 on various cytokines in vivo by staining for IL8/ CXCL1 or their downstream signals molecules in the WT and ybx1 -/- sections. **Corrected**

Reviewer #2 (Remarks to the Author):

The authors report that YBX-1 (also called YB-1) contributes in maintenance of epidermal progenitors via its posttranslational mechanism.
This study may provide some novel information to the related field. However, the manuscript lacks some required data and several critical experiments as indicated in Major comment.

Therefore, I cannot recommend the manuscript for publication in Nature Communication in the present form, because of the high status of the journal.

Major comments

1. YBX-1 has some domains such as alanine-proline rich N-terminal domain, cold shock domain NLS, and C-terminal containing B/A repeats. The authors should determine the functional domains of YB-1 required in the process.

We agree with the reviewer that YBX1 contains well characterized domains and the exploration of their specific function is very valuable. At the same time, we note that, as every other RNA binding protein, its effects are highly context specific and its binding to the targeted transcripts involves not only sequence specificity but also structural changes in the UTRs as well. Therefore a comprehensive analysis of the mode of binding would be outside of the scope of our studies. In line with the reviewer's comment, however, we are presenting now data in the revised Fig. 6C, which point to the essential role of the YBX1 cold shock domain for its modulatory role on cytokine translation in epidermal progenitors.

2. To further demonstrate post-transcriptional function of YB-1 in maintenance of epidermal progenitors, the authors should perform some rescue experiments using wild-type and some mutants YB-1 as characterized above. In the knocking-down experiment, siRNA resistant wild-type and mutants YB-1 should be used. Moreover, in the experiment using knockout mouse, YB-1-adenovirus may be applicable to embryonic epidermis.

We appreciate this concern and as clarified in our response to reviewer 1, we are now presenting several rescue experiments in Fig. 6B and the entire new Fig. 8.

3. To further demonstrate post-transcriptional function of YB-1 in maintenance of epidermal progenitors, the authors should perform quantitative proteome analysis of the protein expression patterns in the above knocking-down/add-back experiment. Moreover, in the experiment using knockout mouse in treatment with YB-1-adenovirus the authors had better perform the proteome analysis.

Proteomic analysis of the YBX1 KD and wild type cells would certainly be a very informative experiment and we have considered several approaches to perform it but were severely limited by the nature of the primary cultures in our study. To execute a robust analysis, we should examine the newly synthesized proteome only but CLICK chemistry is not compatible with the primary human keratinocytes and therefore we believe that focusing on the subset of studied cytokines is the most appropriate approach at this time.

Reviewer #3 (Remarks to the Author):

In their manuscript Kwon and colleagues determine the RNA-binding protein YBX1 as essential regulator of epidermal progenitor cell populations. The authors identify YBX1 in an unbiased screen for RNA-binding proteins in primary human keratinocytes that are down-regulated with terminal differentiation. Deletion of YBX1 in human keratinocytes and the mouse epidermis reduces cell proliferation of progenitor populations but does not affect terminal differentiation. Instead, deletion of YBX1 induces cell cycle arrest and cellular senescence. Mechanistically the authors show that YBX1 suppresses translation of IL-4 and CXCL1 mRNAs, and thereby protects epidermal progenitor cells from cytokine-mediated

senescence.

YBX1 is a multifunctional protein and its up-regulation is found in aggressive tumours. In addition, YBX1 has been implicated in regulating cellular senescence before, yet the underlying mechanism how YBX1 opposes senescence are unknown. Therefore, the manuscript is potentially interesting for the field.

Comments:

1. Figure 2B. As a positive control, involucrin (or any other terminal differentiation factor) expression should be shown.

We agree and have now included the appropriate controls.

2. The authors are very sparse with explaining how they differentiated the keratinocytes. They should elaborate in more detail how the suspension-induced differentiation assay and the spontaneous differentiation assays have been performed.

We included a detailed explanation in the revised methods.

3. Figure 3. Even when the mice have been published already, the author need to provide evidence that YBX1 RNA and protein expression is lost in the knockout mouse epidermis. In addition, the consequences of YBX1-deletion on epidermal progenitor proliferation and differentiation need quantification.

We apologize for this oversight: the controls are included in Fig. S2.

4. Since the authors study a total knockout mouse for YBX1, it is not clear from figure 3 whether the phenotype is caused by a direct role of YBX1 in the epidermis or whether the skin phenotype is rather due to the overall developmental delay of the mouse embryo. At the very minimum, the authors should measure colony forming efficiency of the mouse keratinocytes. It would be also interesting to know whether the IL8 and CXCR1 are conserved targets of YBX1 in the mouse.

We have included the characterization of the effects of YBX1 KO on the production and secretion of senescence associated cytokines in mouse cells in the revised Figs. 5 and 7 and we have provided a detailed explanation in the response to Reviewer 1.

5. Figure 4. Can the effect on proliferation and cell cycle be rescued by an siRNA-insensitive YBX1-construct or treatment with the CXCR2 antagonist?

Rescue experiments are now described in Fig. 8 and SFig. 5

6. The authors conclude from figure 4 that 'These findings indicate that while YBX1 maintains epidermal progenitor homeostasis, it does not directly target the transcriptional control of keratinocytes commitment to differentiation'. Measuring the RNA expression of only three differentiation markers (Fig. 4D) does not warrant such a strong statement. It is not clear to

me why the authors do not provide further evidence for this claim using the RNA seq data obtained from these cells. Also, the RNA seq data are not provided with the manuscript and no link to a public database is provided.

We agree with this comment and have now provided further evidence to this statement from the RNA seq data (Table S3).

7. Figure 6B. The luciferase assay is not convincing given that also the control vector seems to significantly increase in luciferase activity. Why are no data shown for CXCL1 and 2 or the positive control YBX1 itself?

We have changed the experimental system for the promoter assay and are providing new findings in Fig. 6 as well as a detailed explanation in the response to Reviewer 1.

8. Figure 7D needs to be confirmed by Western blot for H3K9me3.

We would like to note that in our system the total amount of H3K9me3 (as detected by Western blotting) is less relevant since differentiating keratinocytes from the upper epidermal layer naturally express it. The identification of H3K9me3 in the basal epidermal progenitors is a prominent marker of epidermal senescence and we are providing now better images with well marked basal layer to document this finding (Fig. 7C).

9. The final conclusion of the authors is that 'YBX1 is able to bind to a subset of cytokine transcripts and prevent their translation thus protecting proliferating epidermal progenitors from undergoing replicative senescence'. This claim is not really supported by data and too strong given that no rescue experiments on keratinocyte proliferation and cell cycle (see also point 5) are provided.

We addressed this concern by multiple rescue experiments as mentioned in the response to Reviewer 1.

10. No table S4 was provided but it is mentioned in the text. Corrected.

Reviewers' Comments:

Reviewer #1:

Remarks to the Author:

The authors have adequately addressed my concerns.

Reviewer #2:

Remarks to the Author:

Major comments

1. YBX-1 has some domains such as alanine-proline rich N-terminal domain, cold shock domain NLS, and C-terminal containing B/A repeats. The authors should determine the functional domains of YB-1 required in the process.

We agree with the reviewer that YBX1 contains well characterized domains and the exploration of their specific function is very valuable. At the same time, we note that, as every other RNA binding protein, its effects are highly context specific and its binding to the targeted transcripts involves not only sequence specificity but also structural changes in the UTRs as well. Therefore a comprehensive analysis of the mode of binding would be outside of the scope of our studies. In line with the reviewer's comment, however, we are presenting now data in the revised Fig. 6C, which point to the essential role of the YBX1 cold shock domain for its modulatory role on cytokine translation in epidermal progenitors.

The authors use YBX1 mutant lacking both N-terminus alanine-proline rich region and cold shock domain in 3'UTR-CXCL1 reporter assay (Fig.6C). This result only indicated that

N-terminus alanine-proline rich region and cold shock domain are required for YBX1 function. They do not identify functional domain of YBX1 which I requested.

2. To further demonstrate post-transcriptional function of YB-1 in maintenance of epidermal progenitors, the authors should perform some rescue experiments using wild-type and some mutants YB-1 as characterized above. In the knocking-down experiment, siRNA resistant wildtype and mutants YB-1 should be used. Moreover, in the experiment using knockout mouse, YB-1-adenovirus

may be applicable to embryonic epidermis.

We appreciate this concern and as clarified in our response to reviewer 1, we are now presenting several rescue experiments in Fig. 6B and the entire new Fig. 8.

Fig.6B?

In Fig.8B, the authors use CXCR2 antagonist for rescue experiment. I agree this use and the result. However, I required different rescue experiment using siRNA resistant wildtype and mutants YBX1 to strengthen the reliability for YBX1-knockdown experiment.

3. To further demonstrate post-transcriptional function of YB-1 in maintenance of epidermal progenitors, the authors should perform quantitative proteome analysis of the protein expression patterns in the above knocking-down/add-back experiment. Moreover, in the experiment using knockout mouse in treatment with YB-1-adenovirus the authors had better perform the proteome analysis.

Proteomic analysis of the YBX1 KD and wild type cells would certainly be a very informative experiment and we have considered several approaches to perform it but were severely limited by the nature of the primary cultures in our study. To execute a robust analysis, we should examine the newly synthesized proteome only but CLICK chemistry is not compatible with the primary human keratinocytes and therefore we believe that focusing on the subset of studied cytokines is the most appropriate approach at this time.

I think that proteomic analysis is necessary to clarify YBX1 function, but it is disappointing that they will not respond to my request.

Infection of YBX1-adenovirus to YBX1 KO mice can rescue the phenotype ?

Reviewer #3:

Remarks to the Author:

The authors have addressed the majority of the referees concerns by either providing additional data or clarifying the text. There are a couple of remaining technical issues, the authors may want to address.

1. Figure S2 now provides the confirmation of the knock-out mice, yet an IF of the embryonic skin is not provided. The authors should provide that staining to show the specificity of the antibody used in Figure 2F,G.

2. The authors now provide quantifications for the observations but it is unclear how the samples have been counted. For example, in figure 2B the label only reads Ki67-positive cells. It is unclear whether this is by image or any other scale has been used. It may be better to provide the percentage of Ki67-positive cells because the overall cellularity of the knockout skin seems reduced in figure 2C,D.

3. Figure 4C: The authors use the cell surface markers CD49 and CD71 to distinguish between cycling and quiescent epidermal progenitor cells. No controls for the FACS are provided (e.g. isotype controls and histograms). It is also unclear, why the samples were pooled instead using them as replicates. The authors conclude that these data confirm a decrease in actively cycling epidermal progenitors and confirm the phenotype. However, according to the FACS results also the number of cells characterized by CD49^{low}/CD71^{low} expression increase by almost up to 5 times. Yet, the authors claim that the differentiated cell populations do not change (Fig. 3C,D; Fig. 4D). What is the CD49^{low}/CD71^{low} cell population representing if not differentiated keratinocytes? Presumably, dead cells have been gated out.

4. The sequencing results are not clear to me. The changes on mRNA level are very modest, yet after 4 days of knock-down one would expect to see even indirect changes caused by the changes in cell division and senescence (Fig. 4A,C; Figure 7).

We would like to thank the reviewers for their constructive comments and suggestions. We have revised the manuscript and are now providing new data to address the concerns brought up by the referees and the editors.

We are including our specific responses to each comment separately below in blue:

Reviewers' comments:

Reviewer #2:

The authors use YBX1 mutant lacking both N-terminus alanine-proline rich region and cold shock domain in 3'UTR-CXCL1 reporter assay (Fig.6C). This result only indicated that N-terminus alanine-proline rich region and cold shock domain are required for YBX1 function. They do not identify functional domain of YBX1 which I requested.

We agree with the reviewer that our analysis does not provide the ultimate identification of the functional YBX1 domain in this system. At the same time we believe that the precise mapping of this domain remains outside of the scope of this initial study and the fact that we have identified the N-terminus and the CSD as required for modulation of CXCL1 translation will justify further, more detailed studies into the molecular mechanism of this signaling pathway.

In Fig.8B, the authors use CXCR2 antagonist for rescue experiment. I agree this use and the result. However, I required different rescue experiment using siRNA resistant wildtype and mutants YBX1 to strengthen the reliability for YBX1-knockdown experiment.

The set up of our experimental system relies predominantly on primary cultures of freshly isolated epidermal progenitors, which offers indeed the key advantage to be able to correlate our findings directly to the human epidermis in vivo but at the same time presents a significant challenge to genetically manipulate the cells. We have established quite stringent conditions of loss-of function experiments using RNAi transfection and gain-of-function modifications using viral vectors but the combination of both results most of the times in high toxicity and very low yield of cells. The rescue experiment suggested by the reviewer requires precisely this approach since expression of exogenous DNA in primary keratinocytes cultures is possible only through viral infection. We were, however, able to reproducibly perform a rescue experiment on the decreased cellular numbers upon YBX1 depletion and are presenting it

now in the new Fig. S5. We believe we were able to show convincingly that an RNAi resistant YBX1 adenovirus is able to rescue the decreased cell counts in the cultures with YBX1 knockdown.

I think that proteomic analysis is necessary to clarify YBX1 function, but it is disappointing that they will not respond to my request.

Infection of YBX1-adenovirus to YBX1 KO mice can rescue the phenotype ?.

We regret that we were not able to satisfy the request for proteomic analysis, which as we mentioned before, we attempted several times but again due to the fragile state of the primary progenitor cultures were not able to apply the SILAC approach.

Reviewer #3

Figure S2 now provides the confirmation of the knock-out mice, yet an IF of the embryonic skin is not provided. The authors should provide that staining to show the specificity of the antibody used in Figure 2F,G.

We agree with the reviewer and are now providing in Fig.S2E the immunofluorescent labeling of the wild type and knockout epidermis showing the lack of signal in the YBX1^{-/-} samples.

The authors now provide quantifications for the observations but it is unclear how the samples have been counted. For example, in figure 2B the label only reads Ki67-positive cells. It is unclear whether this is by image or any other scale has been used. It may be better to provide the percentage of Ki67-positive cells because the overall cellularity of the knockout skin seems reduced in figure 2C,D.

We apologize for the oversight noted by the reviewer: the graph shown in Fig. 3B represents percent of Ki67 positive cells calculated out of total KRT5/DAPI positive basal cells. This is now clearly indicated in the Figure and the legend.

Figure 4C: The authors use the cell surface markers CD49 and CD71 to distinguish between cycling and quiescent epidermal progenitor cells. No controls for the FACS are provided (e.g. isotype controls and histograms). It also unclear, why the samples were pooled instead using them as replicates. The authors conclude that these data confirm a decrease in actively cycling epidermal progenitors and confirm the phenotype. However, according to the FACS results also the number of cells characterized by CD49^{low}/CD71^{low} expression increase by almost up to 5 times. Yet, the authors claim that the differentiated cell populations do not change (Fig. 3C,D; Fig. 4D). What is the CD49^{low}/CD71^{low} cell population representing if not differentiated keratinocytes? Presumably, dead cells have been gated out.

We are providing here below the histograms obtained from the same FACS analysis (for Reviewer's reference). As it is evident from these plots, the main outcome of YBX1 depletion in the primary epidermal progenitors is the significant decrease of CD71 labeling and much less of the ITGA6 signal. Since CD71 is a prominent marker of proliferating cells, our data are in agreement with the rest of the findings that YBX1 down modulation results in growth arrest

and inability to proliferate, which among others is a hallmark of cellular senescence. The minimal changes of the ITGA6 signal also correlates with the lack of changes in the differentiation potential of the cells upon YBX1 depletion. We have now included this explanation in the text of the manuscript.

The sequencing results are not clear to me. The changes on mRNA level are very modest, yet after 4 days of knock-down one would expect to see even indirect changes caused by the changes in cell division and senescence (Fig. 4A,C; Figure 7).

We regret that we did not explain well our approach. We performed the RNA-seq analysis at day four after RNAi transfection (as well as the majority of the experiments in this study) since YBX1 is a quite stable protein and its mRNA has to be down-modulated for quite a long time in order to effectively result in protein decrease. We have performed multiple time courses and realized that day 4 is the earliest time point when we can observe YBX1 knock-down. We are providing here an example of such experiment. Therefore we think that the changes observed in the RNA-seq data are mostly direct. In addition we would like to note that relatively modest changes in polysomal mRNA levels are known to result in quite significant protein changes since only the actively translating part of the total mRNA is being analyzed.

Reviewers' Comments:

Reviewer #3:

Remarks to the Author:

The authors addressed all my remaining concerns either by providing the data or through clarifications in the text.